# Safeguarded Stochastic Polyak Step-Sizes for Non-smooth Optimization: Robust Performance Without Small (Sub)Gradients

## Abstract

The stochastic Polyak step size (SPS) has proven to be a promising choice for stochastic gradient descent (SGD), delivering competitive performance relative to state-of-the-art methods on smooth convex and non-convex optimization problems, including deep neural network training. However, extensions of this approach to non-smooth settings remain in their early stages, often relying on interpolation assumptions or requiring knowledge of the optimal solution. In this work, we propose a novel SPS variant — Safeguarded SPS ($\text{SPS}_{safe}$) — for the stochastic subgradient method, and provide rigorous convergence guarantees for non-smooth convex optimization with no need for strong assumptions. We further incorporate momentum into the update rule, yielding equally tight theoretical results. Comprehensive experiments on convex benchmarks and deep neural networks corroborate our theory: the proposed step size accelerates convergence, reduces variance, and consistently outperforms existing adaptive baselines. Finally, in the context of deep neural network training, our method demonstrates robust performance by addressing the vanishing gradient problem.

## 1 Introduction

Adaptive optimization methods have become fundamental tools in machine learning, offering robustness and eliminating the need for manual learning rate tuning. Among the most prominent are AdaGrad and Adam. AdaGrad (Duchi et al., 2011) adapts the learning rate individually for each parameter by accumulating past squared gradients, making it well-suited for sparse features but often suffering from decreasing learning rates over time. Adam (Kingma & Ba, 2015) extends this idea by maintaining exponential moving averages of both the gradient and its squared values, and correcting for initialization bias. As a result, Adam has demonstrated strong empirical performance and has become a standard optimizer in deep learning applications due to its stability and ease of use (Vaswani et al., 2017; Guo et al., 2022; Peebles & Xie, 2023).

In a different direction, adaptive optimization algorithms using Polyak-type step sizes have started gaining recognition for their simplicity and strong practical performance. Unlike traditional adaptive methods that rely solely on gradient information, these approaches determine the learning rate using function values. The classical Polyak step size (PS), originally introduced by Polyak (1987), was proposed as an efficient rule for step size selection in gradient descent for solving convex optimization problems. Although rooted in early optimization literature, these ideas have recently seen a resurgence, particularly in machine learning applications. The Polyak step size was recently extended to stochastic settings. Loizou et al. (2021) proposed and analyzed stochastic gradient descent (SGD) with Stochastic Polyak Step size (SPS) and demonstrated convergence guarantees for convex and non-convex problems while retaining the simplicity of the original rule. The proposed SPS comes with strong convergence guarantees and competitive performance in training DNNs, and it is particularly useful when training over-parameterized models (Loizou et al., 2021). In the last few years, many other works have explored the use of stochastic Polyak step sizes in different training algorithms, including SGD (Garrigos et al., 2023; Orvieto et al., 2022; Jiang & Stich, 2023; Gower et al., 2025), Stochastic Mirror Descent D'Orazio et al. (2023) Local SGD Mukherjee et al. (2024), and SGD with Momentum Wang et al. (2023); Schaipp et al. (2024); Oikonomou & Loizou (2025); Gower et al. (2025). Nevertheless, nearly all existing analyses assume *convexity and smoothness*

| Work | Step Size | No Interpolation? | No $f_i(x^*)$? | Rate |
|------|-----------|-------------------|----------------|------|
| *Stochastic Subgradient Method* | | | | |
| Loizou et al. (2021) | $\gamma_t = \frac{f_i(x^t) - f_i^*}{\|g_i^t\|^2}$ | ✗ | ✓ | $\mathcal{O}(1/\sqrt{T})$ |
| Garrigos et al. (2023) | $\gamma_t = \frac{[f_i(x^t) - f_i(x^*)]_+}{\|g_i^t\|^2}$ | ✓ | ✗ | $\mathcal{O}(1/\sqrt{T})$ |
| Theorem 3.1 | $\gamma_t = \frac{f_i(x^t) - \ell_i^*}{\max\{\|g_i^t\|^2, M\}}$ | ✓ | ✓ | $\mathcal{O}(1/\sqrt{T} + \sigma^2)$ |
| *IMA (Momentum)* | | | | |
| Gower et al. (2025) | $\eta_t = \frac{[f_i(x^t) - f_i(x^*) + \lambda_t \langle g_i^t, x^t - x^{t-1} \rangle]_+}{\|g_i^t\|^2}$ | ✓ | ✗ | $\mathcal{O}(1/\sqrt{T})$ |
| Theorem 3.5, Theorem 3.4 | $\eta_t = \frac{[f_i(x^t) - \ell_i^* + \lambda_t \langle g_i^t, x^t - x^{t-1} \rangle]_+}{\max\{\|g_i^t\|^2, M\}}$ | ✓ | ✓ | $\mathcal{O}(1/\sqrt{T} + \sigma^2)$ |

Table 1: Overview of methods with Polyak-type step sizes analyzed in the convex non-smooth stochastic setting. For every method we list the explicit rule, indicate whether the theory (i) holds *without* the interpolation assumption and (ii) avoids the oracle values $f_i(x^*)$, and report the proven convergence rate on convex–Lipschitz objectives. Rows shaded in green are the new contributions of this work. The constant $M$ in our step sizes is the safeguard constant, for more details see Section 2 and the variance $\sigma^2$ is defined in (2).

while robust guarantees in the non-smooth regime remain scarce. *The understanding of efficient stochastic Polyak-type step sizes in the arguably more challenging non-smooth regime is precisely the main focus of our work.*

**Problem Setup and Main Algorithms.** We focus on the unconstrained finite–sum optimization problem

$$\min_{x \in \mathbb{R}^d} \left[ f(x) = \frac{1}{n} \sum_{i=1}^{n} f_i(x) \right], \tag{1}$$

where each component function $f_i : \mathbb{R}^d \to \mathbb{R}$ is *convex, Lipschitz and **non-smooth*** as well as lower bounded by $\ell_i^*$. Let $X^*$ denote the set of minimizers of (1). We assume that $X^* \neq \emptyset$. This problem is the cornerstone of many machine learning tasks, (Hastie et al., 2009), where the vector $x$ represents the model parameters, $f_i(x)$ is the loss related to the training point $i$, and the goal is to minimize the empirical risk $f(x)$ across all training points.

Two widely used algorithms for solving stochastic, non-smooth convex optimization problems of the form (1) are (i) the *Stochastic Subgradient Method* (SSM) and (ii) the more recent *Iterate Moving Average* (IMA), an equivalent algorithm to the stochastic subgradient method with momentum (Sebbouh et al., 2021).

SSM, tracing back to the seminal work of Robbins & Monro (1951) and later formalized for convex objectives by Shor (1985) and Nedić & Bertsekas (2001). It has the following update rule

$$x^{t+1} = x^t - \gamma_t g_i^t, \tag{SSM}$$

where $g_i^t \in \partial f_i(x^t)$ is the stochastic subgradient, $i$ is uniformly drawn from $\{1, \ldots, n\}$, and a $\gamma_t > 0$ is the step size of the method.

IMA extends SSM by adding a new iterate $(z^t)$: Each iteration first performs a subgradient step on $z^t$ and then averages the result with the previous iterate $x^t$. The update rule is given by:

$$z^{t+1} = z^t - \eta_t g_i^t, \qquad \text{where} \qquad g_i^t \in \partial f_i(x^t)$$

$$x^{t+1} = \frac{\lambda_{t+1}}{\lambda_{t+1} + 1} x^t + \frac{1}{\lambda_{t+1} + 1} z^{t+1}. \tag{IMA}$$

Defazio & Gower (2021) show that this two-sequence scheme is *algebraically equivalent* to the more familiar Stochastic Heavy Ball (SHB) momentum update rule $x^{t+1} = x^t - \hat{\gamma}_t g_i^t + \beta(x^t - x^{t-1})$ with $1 + \lambda_{t+1} = \lambda_t \beta_t$ and $\eta_t = (1 + \lambda_{t+1})\hat{\gamma}_t$, (Ma & Yarats, 2019; Kidambi et al., 2018; Liu et al., 2020; Sebbouh et al., 2021; Oikonomou & Loizou, 2025).

In our work, we focus on adaptive variants of both SSM and IMA, and provide Polyak-type step-sizes $\gamma_t$ for solving convex non-smooth optimization problems.

**Prior non-smooth Polyak-type Results.** For the Stochastic Subgradient Method (SSM) on convex and Lipschitz objectives, Loizou et al. (2021) proved that SSM with the step size $\gamma_t = \frac{f_i(x^t) - f_i^*}{\|g_i^t\|^2}$,

where $f_i^* = \inf_{x \in \mathbb{R}^d} f_i(x)$, converges at a rate of $\mathcal{O}(T^{-1/2})$ *only* under the strong *interpolation* assumption (i.e. there exists $x^* \in X^*$ such that $f_i(x^*) = f_i^*$ for all $i$). Garrigos et al. (2023) later proposed the step size

$$\gamma_t = \frac{[f_i(x^t) - f_i(x^*)]_+}{\|g_i^t\|^2}, \tag{SPS$^*$}$$

where $[z]_+ = \max\{z, 0\}$, obtaining the same $\mathcal{O}(T^{-1/2})$ bound without interpolation, but at the cost of requiring knowledge of each individual optimal loss value (i.e., $f_i(x^*)$). In the momentum setting, Gower et al. (2025) extended this idea to the Iterate Moving Average (IMA) update rule, showing that the IMA method with the step size

$$\eta_t = \frac{[f_i(x^t) - f_i(x^*) + \lambda_t \langle g_i^t, x^t - x^{t-1} \rangle]_+}{\|g_i^t\|^2}, \tag{IMA-SPS}$$

achieves the same rate for both the Cesàro average and the last iterate, again assuming access to $f_i(x^*)$. For a summary of these results, we refer to Table 1.

**Limitations and our remedy.**  Taken together, the existing variants of Polyak-type algorithms either (i) converge only under interpolation, or (ii) rely on oracle information such as $f_i(x^*)$ that is unavailable in practice. The step sizes we introduce in this work eliminate both drawbacks: they are *fully adaptive*, i.e, need no additional problem knowledge, and match the $\mathcal{O}(T^{-1/2})$ rate for convex, Lipschitz objectives (up to a neighborhood) without extra information, in both plain SSM and its momentum variant IMA.

## 1.1 MAIN CONTRIBUTIONS

Our main contributions are summarized below:

⋄ **Safeguarded Polyak Step size for SSM.** We design a new Polyak-type rule, named safeguarded SPS ($\text{SPS}_{safe}$) for SSM, that does not allow the subgradient used in the stochastic Polyak step size to become small. This single safeguard removes the need for any oracle information (e.g. $f_i(x^*)$) and attains $\mathcal{O}(T^{-1/2})$ convergence a neighborhood of solution, for stochastic, convex & Lipschitz objectives *without* the interpolation assumption.

⋄ **An in-depth understanding of $\text{SPS}_{safe}$.** We explain the benefits of $\text{SPS}_{safe}$ in terms of theory and experiments compared to prior works on Polyak-type step sizes. The proposed rule is the first Polyak-type step size for SSM that remains *genuinely adaptive*: it never becomes a constant update, irrespective of the chosen safeguard threshold. Earlier variants (Loizou et al., 2021; Wang et al., 2023; Zhang et al., 2025) can yield a *constant* step size once a user-specified upper bound on the step size is small enough. In addition, we establish a connection between the proposed step size rule and the clipping mechanism used extensively in modern DNN training. The safeguarding mechanism can be interpreted as an *in-step* gradient-clipping operation, thereby supplying the first theoretical guarantees for Polyak-style *clipped* SSM, an update rule widely used to mitigate both exploding and vanishing gradients in DNNs.

⋄ **Safeguarded Polyak-type step size for IMA and last-iterate convergence.** We extend the safeguarded step size ideas to the Iterate Moving Average (IMA) update rule, yielding the Safeguarded Polyak-type step size $\text{IMA-SPS}_{safe}$. For this step size selection, we prove $\mathcal{O}(T^{-1/2})$ convergence for IMA in terms of *both* the Cesàro average and the last-iterate. Our proposed analysis provides the first convergence guarantees for an adaptive momentum method (through the equivalence of IMA and SSM with heavy ball momentum) that does not require any strong assumption (e.g., the knowledge of $f_i(x^*)$, (Gower et al., 2025)).

⋄ **Numerical Evaluation.**  In Section 4, we present extensive experiments validating different aspects of our theoretical results (sensitivity analysis of our step size and comparison with other Polyak step sizes in the non-smooth setting). We also assess the performance of $\text{SPS}_{safe}$ and $\text{IMA-SPS}_{safe}$ in training deep neural networks for multi-class image classification problems and compute the gradient norms under $\text{SPS}_{safe}$, confirming they do not collapse to small values. Reproducible code is provided with the submission.

## 2 SAFEGUARDED STOCHASTIC POLYAK STEP SIZES

This section introduces the *Safeguarded* Polyak step sizes for SSM and IMA and we explain how our theory differs from previous works. Next, we show how the safeguarded step sizes improves practical behaviour in deep-network training. Finally, we show that the safeguard can be interpreted as an adaptive form of gradient clipping, thereby combining Polyak updates with clipped-SSM techniques.

**SPS$_{safe}$ for the Stochastic Subgradient Method.** To stabilise Polyak step s in the non-smooth setting we introduce

$$\gamma_t = \frac{f_i(x^t) - \ell_i^*}{\max\{\|g_i^t\|^2, M\}}, \qquad \text{(SPS}_{safe}\text{)}$$

where $g_i^t \in \partial f_i(x^t)$ and $M > 0$ is a user–chosen safeguard. The $\max\{\cdot, M\}$ term prevents the denominator from approaching zero, thereby avoiding the *exploding step size* problem that arises when $\|g_i^t\| \to 0$ in deep neural networks. Earlier work of Loizou et al. (2021) controlled the same phenomenon by clipping the *whole* polyak step size, taking $\gamma_t = \min\{\text{Polyak step}, \gamma_b\}$ with a fixed upper bound $\gamma_b$. By contrast, SPS$_{safe}$ keeps the numerator intact and instead boosts very small gradients, which empirically produces smoother step sizes and tight theoretical bounds (see Theorem 3.1).

In essence, the single hyper-parameter $M$ replaces both the clipping constant $\gamma_b$ of SPS$_{max}$ and the oracle values $f_i(x^*)$ required by SPS*/IMA-SPS, yielding a fully adaptive, practically parameter-free step-size family for non-smooth optimization.

**IMA-SPS$_{safe}$ for momentum.** The Iterate-Moving-Average (IMA) framework of Gower et al. (2025) selects $\eta_t = [f_i(x^t) - f_i(x^*) + \lambda_t\langle g_i^t, x^t - x^{t-1}\rangle]_+/\|g_i^t\|^2$, but requires the unknown optimal loss $f_i(x^*)$. We remove this oracle dependence and simultaneously safeguard against vanishing gradients with

$$\eta_t = \frac{[f_i(x^t) - \ell_i^* + \lambda_t\langle g_i^t, x^t - x^{t-1}\rangle]_+}{\max\{\|g_i^t\|^2, M\}}, \qquad \text{(IMA-SPS}_{safe}\text{)}$$

where $\lambda_t \geq 0$ is the usual IMA momentum parameter. When $\lambda_t = 0$ this reduces to SPS$_{safe}$, while for $\lambda_t > 0$ it becomes a safeguarded analogue of stochastic heavy ball that enjoys last-iterate and Cesàro guarantees (Theorems 3.4 and 3.5) without any knowledge of $f_i(x^*)$.

### 2.1 THEORETICAL BENEFIT

To the best of our knowledge, the literature contains only two theoretical guarantees for Polyak-type step sizes in the convex–Lipschitz regime (non-smooth regime) (Loizou et al., 2021; Garrigos et al., 2023), both of which require strong, often impractical, assumptions.

Let $f_i$ be convex and $G$-Lipschitz functions and let $\overline{x}^T = \frac{1}{n}\sum_{t=0}^{T-1} x^t$. Then the two papers provide the below convergence guarantees:

- (Loizou et al., 2021): Assume that *interpolation* condition holds. Consider the iterates of SSM with the step size given by $\gamma_t = \frac{f_i(x^t) - f_i^*}{\|g_i^t\|^2}$. Then $\mathbb{E}[f(\overline{x}^T) - f(x^*)] \leq \frac{G\|x^0 - x^*\|}{\sqrt{T}}$.
- (Garrigos et al., 2023): Consider the iterates of SSM with the step size given by $\gamma_t = \frac{[f_i(x^t) - f_i(x^*)]_+}{\|g_i^t\|^2}$. Then $\mathbb{E}[f(\overline{x}^T) - f(x^*)] \leq \frac{G\|x^0 - x^*\|}{\sqrt{T}}$. The use of $[z]_+$ is needed to enforce the step size to be positive.

Both guarantees therefore rely on conditions seldom met in practice: either exact interpolation or full knowledge of $f_i(x^*)$. Note also that when interpolation is assumed, $f_i^* = f_i(x^*)$, making the step size in both papers concise. Our main theorem (Theorem 3.1) closes this gap. Using the safeguarded step size SPS$_{safe}$, we achieve the same $\mathcal{O}(T^{-1/2})$ convergence to a neighborhood of the solution, and the results hold *without* assuming interpolation and *without* oracle information. The final bound has the familiar smooth-setting structure: it converges to a neighborhood whose radius scales with the gradient variance and collapses to zero in the interpolated case, thereby recovering Loizou et al. (2021) as a special instance.

## 2.2 PRACTICAL CONSIDERATIONS IN TRAINING OF DNNs

**How often is a Polyak rule actually used?** Polyak step sizes are praised for being *adaptive*, yet in deep neural networks experiments, they can end up acting like fixed learning rates. To investigate this phenomenon we run ResNet-20 (He et al., 2016) on CIFAR-10 dataset (Krizhevsky et al., 2009). Recall, Loizou et al. (2021) proposed the following step size

$$\gamma_t^{\text{SPS}_{\max}} = \min\left\{\gamma_t^{\text{SPS}}, \gamma_b\right\}, \quad \gamma_t^{\text{SPS}} = \frac{f_i(x^t) - \ell_i^*}{c\|g_i^t\|^2}. \tag{SPS$_{\max}$}$$

For SPS$_{\max}$, we swept the constant $c$ over $c \in \{0.1, 0.2, \ldots, 1.0\}$, set $\gamma_b = 1$ and $\ell_i^* = 0$, and trained for 100 epochs. The best accuracy, which reached 87.88%, occurred at $c = 0.4$. However, a counter revealed that in **31.8%** of the iterations the algorithm selected the constant value $\gamma_b$ rather than the "true" Polyak step $\gamma_t^{\text{SPS}}$. Thus almost one-third of the updates were effectively constant.

In order to increase performance, Loizou et al. (2021) use the following smoothing rule for the clipping hyper-parameter[1] $\gamma_b^t = \tau^{b/n}\gamma_{t-1}$ with $\tau = 2$, batch size $b$, and dataset size $n$. The step size then takes the following form:

$$\gamma_t^{\text{Smooth SPS}_{\max}} = \min\left\{\gamma_t^{\text{SPS}}, \gamma_b^t\right\} = \min\left\{\gamma_t^{\text{SPS}}, \tau^{b/n}\gamma_{t-1}^{\text{Smooth SPS}_{\max}}\right\}. \tag{Smooth SPS$_{\max}$}$$

Adopting this rule and retuning $c$ over $c \in \{0.1, 0.2, \ldots, 1.0\}$, lifts accuracy to 89.79% for $c = 0.5$, but at a hidden cost: **98.45%** of the step s now use $\gamma_t = \gamma_b^t$. In practice, the method behaves almost like a decreasing learning-rate scheme, the step size is plotted in Figure 1.

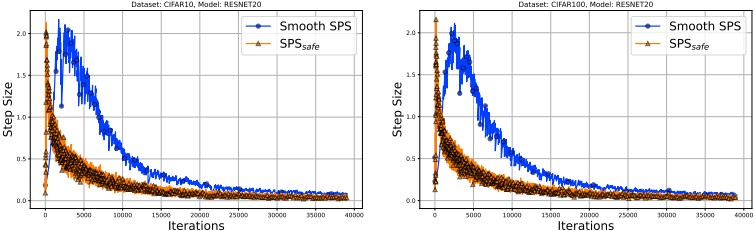

Figure 1: Comparison of Smooth SPS$_{\max}$ and SPS$_{safe}$ in the training of ResNet20 in CIFAR-10 (left plot) and CIFAR-100 (right plot).

**Safeguarded Polyak.** Replacing SPS$_{\max}$ with our safeguarded step SPS$_{safe}$ yields 90.39% accuracy, competitive with the smoothed baseline, while *never* collapsing to a constant value. SPS$_{safe}$ also exhibits a smoothing behaviour similar to Smooth SPS$_{\max}$, see Figure 1 for a direct comparison between the two step sizes. Crucially, this behaviour is backed by explicit convergence guarantees, whereas the smoothed heuristic offers none.

## 2.3 CONNECTIONS WITH CLIPPING: ADAPTIVE CLIPPED SSM VIA SPS$_{safe}$

A convenient way to stabilise stochastic subgradient methods is to clip each individual gradient before applying the update. Given a threshold $c > 0$, define the clipping operator

$$\text{clip}_c(g) := \min\left\{1, \frac{c}{\|g\|}\right\}g,$$

which leaves gradients with $\|g\| \leq c$ unchanged and rescales larger ones to have norm exactly $c$. The *clipped* stochastic subgradient method (clipped SSM) therefore has the following update rule:

$$x^{t+1} = x^t - \tilde{\gamma}_t\text{clip}_c\left(g_i^t\right) = x^t - \tilde{\gamma}_t\min\left\{1, \frac{c}{\|g_i^t\|}\right\}g_i^t, \tag{Clipped SSM}$$

with step size $\tilde{\gamma}_t > 0$. By capping overly large updates and boosting very small ones, clipping simultaneously protects against exploding gradients and mitigates the vanishing-gradient phenomenon.

The following proposition explained how SPS$_{safe}$ can be modified for Clipped SSM, thus providing an adaptive step size for Clipped SSM. For the proof we refer to the appendix.

---

[1]Similar smoothing procedures have been used in Tan et al. (2016); Vaswani et al. (2019)

**Proposition 2.1.** *SSM with $SPS_{safe}$ and $M = c^2$ is algebraically equivalent to Clipped SSM with the adaptive step size $\tilde{\gamma}_t = \frac{f_i(x^t) - \ell_i^*}{c \max\{c, \|g_i^t\|\}}$.*

**Small subgradients (vanishing gradient phenomenon).** Gradient clipping augments SSM with a simple safety mechanism: whenever a stochastic gradient's norm falls outside a user-set bound, it is rescaled. This single mechanism tackles both extremes of training instability. By capping very large updates, it prevents the exploding gradient spikes that can derail optimization, while simultaneously boosting tiny gradients to a usable magnitude, it mitigates the 'vanishing-gradient' stall common in deep neural networks. In the same spirit, the safeguard in our step-size formula $SPS_{safe}$ prevents the learning rate from blowing up by effectively normalising updates when subgradients become extremely small. There is a growing deterministic literature that combines Polyak steps with Clipped SSM in the non-smooth setting, notably the recent analyses of Gorbunov et al. (2025) and Takezawa et al. (2024). These results, however, target $(L_0, L_1)$-smooth objectives, whereas we focus on globally Lipschitz, but otherwise non-smooth, functions and operates in the *stochastic* regime.

## 3 CONVERGENCE ANALYSIS

This section states the convergence guarantees for $SPS_{safe}$ and $IMA\text{-}SPS_{safe}$, full proofs are deferred to the appendix. Throughout, each loss $f_i$ is convex and $G$–Lipschitz.

**Variance measure.** To quantify gradient noise in the non-smooth setting we use

$$\sigma^2 := \left( \mathbb{E}_i \left[ (f_i(x^*) - \ell_i^*)^2 \right] \right)^{\frac{1}{2}}. \tag{2}$$

The standard definition of the variance in the Stochastic Polyak step sizes literature Loizou et al. (2021); Oikonomou & Loizou (2025); Wang et al. (2023); Zhang et al. (2025) is given by $\hat{\sigma}^2 := \mathbb{E}_i \left[ f_i(x^*) - \ell_i^* \right]$. Note that by Jensen's inequality we have that $\hat{\sigma}^2 \leq \sigma^2$. Moreover, $\sigma^2 < \infty$ whenever each $f_i$ is lower-bounded. When problem (1) is interpolated, i.e. there exists $x^* \in X^*$ such that $f_i(x^*) = f_i^*$, then choosing $\ell_i^* = f_i^*$ we get $\sigma^2 = 0$. Many modern machine learning models satisfy this condition. Examples include non-parametric regression (Liang & Rakhlin, 2020) and over-parameterized deep neural networks (Zhang et al., 2021; Ma et al., 2018).

### 3.1 STOCHASTIC SUBGRADIENT METHOD

Firstly, we focus on the stochastic subgradient method, where we have the following theorem.

**Theorem 3.1.** *Consider the iterates of SSM with the step size ($SPS_{safe}$). Then*

$$\mathbb{E}[f(\overline{x}^T) - f(x^*)] \leq \frac{\sqrt{\max\{G^2, M\}}\|x^0 - x^*\|}{\sqrt{T}} + \sqrt{\frac{\max\{G^2, M\}}{M}}\sigma^2,$$

*where $\overline{x}^T = \frac{1}{T} \sum_{t=0}^{T-1} x^t$.*

Theorem 3.1 eliminates two issues that limit the best previous bounds of Loizou et al. (2021) and Garrigos et al. (2023): it needs neither the interpolation condition nor oracle access to the values $f_i(x^*)$, and still achieves the rate $\mathcal{O}(T^{-1/2})$ to a neighborhood of the solution.

Because the safeguard acts solely through the denominator, the same result—via the equivalence in Proposition 2.1—yields the first stochastic guarantee for the clipped-SSM update (Clipped SSM). Furthermore, via the equivalency established in Proposition 2.1, this theorem provides convergence guarantees for Clipped SSM. On a different note, the coefficient of $\sigma^2$ is decreasing with respect to $M$, a trend we confirm empirically in Section 4.1. However, a limitation remains: the radius of that neighborhood is independent of $\gamma_t$, so a simple decreasing step size cannot force convergence to the exact optimum when gradient noise is present.

Now consider two notable specializations of Theorem 3.1. First, in the *interpolated* regime, where every sample can be fitted exactly, we choose the lower bounds $\ell_i^* = f_i^*$. The variance term then disappears and we obtain an exact convergence rate.

**Corollary 3.2** (Interpolation). *Under interpolation with $\ell_i^* = f_i^*$,*

$$\mathbb{E}[f(\overline{x}^T) - f(x^*)] \leq \frac{\sqrt{\max\{G^2, M\}}\|x^0 - x^*\|}{\sqrt{T}}.$$

When $M = 0$ this reproduces the step size and rate of Loizou et al. (2021), as seen in Section 2.1, thus recovering earlier results as a special case of the safeguarded framework. Next, we concentrate on the deterministic or full batch regime.

**Corollary 3.3** (Deterministic SSM). *In the deterministic regime, we have $\sigma^2 = 0$, so Theorem 3.1 suggests*

$$\min_{t \in [T]} \{f(x^t) - f(x^*)\} \leq \frac{\sqrt{\max\{G^2, M\}}\|x^0 - x^*\|}{\sqrt{T}}.$$

There is a plethora of Polyak step size analyses in the determinstic regime assuming non-smoothness. Polyak's seminal work in Polyak (1964) on deterministic subgradient descent with the Polyak step size already established an $\mathcal{O}(G\|x^0 - x^*\|/\sqrt{T})$ bound for nonsmooth convex objectives. Subsequent studies strengthened the guarantees under additional structure: Davis et al. (2018) proved *linear* convergence for the same step size when the objective is weakly convex and sharp, while Hazan & Kakade (2019) obtained an $\mathcal{O}(1/T)$ rate for strongly convex, Lipschitz functions.

## 3.2 ITERATE MOVING AVERAGE (MOMENTUM)

In this section we focus on IMA.

**Theorem 3.4.** *Consider the iterates of IMA with the step size (IMA-SPS$_{safe}$) and let $(\lambda_t)_{t>0}$ be a decreasing sequence of nonnegative reals. Then*

$$\mathbb{E}[f(\overline{x}^T) - f(x^*)] + \sum_{t=0}^{T-1} \frac{\lambda_t}{T} \mathbb{E}[B_f(x^{t-1}, x^t)] \leq \frac{\sqrt{\max\{G^2, M\}}\|x^0 - x^*\|}{\sqrt{T}} + \sqrt{\frac{\max\{G^2, M\}}{M}}\sigma^2,$$

*where $\overline{x}^T = \frac{1}{T}\sum_{t=0}^{T-1} x^t$ and $B_f(x, y) = f(x) - f(y) - \langle \partial f(y), x - y \rangle$ is the Bregman divergence.*

The bound mirrors that of Theorem 3.1, but with an additional non-negative Bregman divergence term. The most common scenario for the sequence $\lambda_t$ is being held fixed ($\lambda_t = \lambda$). When $\lambda = 0$ we recover exactly Theorem 3.1. However, when $\lambda > 0$ we have an extra non-negative term on the left hand side. This suggests, but does not force, a speed-up over the plain subgradient method.

So far we have only provided guarantees for the Cesaro average. In the next theorem we prove convergence of the last iterate.

**Theorem 3.5.** *Consider the iterates of IMA with the step size (IMA-SPS$_{safe}$) and let $\lambda_t = t$. Then*

$$\mathbb{E}[f(x^{T-1}) - f(x^*)] + \frac{1}{T}\sum_{t=0}^{T-1} t\,\mathbb{E}[B_f(x^{t-1}, x^t)] \leq \frac{\sqrt{\max\{G^2, M\}}\|x^0 - x^*\|}{\sqrt{T}} + \sqrt{\frac{\max\{G^2, M\}}{M}}\sigma^2.$$

This result provides an explicit guarantee for the *last* iterate, often the quantity of practical interest, while retaining the same rate as the Cesàro bound. Similar remarks as in the previous section hold about the neighborhood in this regime, namely the neighborhood is decreasing with respect to safeguard $M$ (see also Section 4.1).

Fewer results exist for Polyak-type step sizes paired with momentum than for their momentum-free counterparts. Wang et al. (2023), treats a heavy-ball variant under *smooth* convex losses and achieves an $\mathcal{O}(1/T)$ rate. Oikonomou & Loizou (2025) study the Stochastic Heavy Ball momentum via IMA, again assuming smooth objectives. The only work that drops smoothness is the analysis of Gower et al. (2025), which attains the $\mathcal{O}(T^{-1/2})$ rate in the convex, Lipschitz setting, but at the cost of requiring the quantities $f_i(x^*)$. Our safeguarded momentum rule removes this oracle dependence while preserving the same rate.

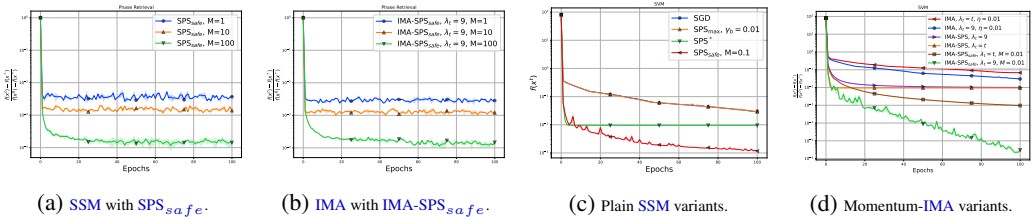

(a) SSM with SPS$_{safe}$.  (b) IMA with IMA-SPS$_{safe}$.  (c) Plain SSM variants.  (d) Momentum-IMA variants.

Figure 2: Sensitivity analysis of the safeguarded Polyak step size to the threshold $M$ (Panels a-b for Phase Retrieval) and comparison against SPS variants (Panels c–d for SVM).

## 4 NUMERICAL EXPERIMENTS

We now examine the empirical behaviour of the safeguarded step sizes SPS$_{safe}$ and IMA-SPS$_{safe}$. The first series of experiments targets convex, *non-smooth* objectives and is designed to validate the theory developed in Section 3. The second series moves to deep-learning benchmarks, measuring the impact of the step sizes on classification accuracy.

### 4.1 VALIDATION OF THE THEORY ON SVMS AND PHASE RETRIEVAL

In this part, we empirically validate our theoretical results and illustrate the main properties of SPS$_{safe}$ and IMA-SPS$_{safe}$ that our theory suggests in Section 3. In these experiments, we focus on Support Vector Machines (SVMs) and Phase Retrieval problems and we evaluate the performance of our step sizes on synthetic data. Recall that we want to solve the problem $\min_{x \in \mathbb{R}^d} f(x) = \frac{1}{n}\sum_{i=1}^{n} f_i(x)$. Let $A \in \mathbb{R}^{n \times d}$ denote the feature matrix and $b \in \mathbb{R}^n$ the labels.

**SVM.** The individual loss and subgradient is given by

$$f_i(x) = \left| \langle A_i, x \rangle^2 - b_i \right|, \qquad \partial f_i(x) = 2\langle A_i, x \rangle \mathrm{sgn}\left( \langle A_i, x \rangle - b_i \right) A_i,$$

where $\mathrm{sgn}(\cdot)$ denotes the sign function.

**Phase Retrieval.** The individual loss and subgradient is given by

$$f_i(x) = \max\left(0, 1 - b_i \langle A_i, x \rangle\right), \qquad \partial f_i(x) = -\delta_{b_i \langle A_i, x \rangle \le 1} b_i A_i,$$

where $\delta_X = 1$ if condition $X$ holds, otherwise $\delta_x = 0$.

**Sensitivity to the safeguard $M$.** We study how the choice of the threshold $M$ influences both the plain and momentum variants of our proposed step sizes. The experiment is a synthetic phase-retrieval task with $n = 300$ samples and dimension $d = 10$; rows of $A$ and the vector $b$ are drawn i.i.d. from $\mathcal{N}(0, 1)$. We run SSM equipped with SPS$_{safe}$ and IMA equipped with IMA-SPS$_{safe}$ for 100 epochs, using a batch size of $n/10 = 30$. Three values of the safeguard are tested, $M \in \{1, 10, 100\}$, and for the momentum experiment, we set $\lambda_t = 9$, which is equivalent to the heavy-ball parameter $\beta = 0.9$. Each configuration is averaged over three independent trials; mean curves and one-standard-deviation bands are reported in Figures 2a and 2b. Consistently with the bounds of Theorems 3.1 and 3.4, a larger $M$ tightens the neighborhood: both algorithms converge to progressively lower error plateaus as the safeguard increases.

**Comparison with existing Polyak step sizes.** We next benchmark the safeguarded rules against their best-tuned classical counterparts. The task is a synthetic SVM with $n = 300$ samples and dimension $d = 100$; both the feature matrix $A$ and the label vector $b$ are drawn from $\mathcal{N}(0, 1)$ as in the previous section. For SPS$_{safe}$ and SPS$_{safe}$ we sweep the safeguard over $M \in \{0.01, 0.1, 1.0, 1.0, 10.0, 100.0\}$. The plain SSM is tuned over four learning rates $\gamma \in \{10^{-4}, 10^{-3}, 10^{-2}, 10^{-1}\}$, while the constant step size IMA baseline is tuned over $\eta \in \{10^{-4}, 10^{-3}, 10^{-2}, 10^{-1}\}$ for two momentum choices: $\lambda_t = 9$ (equivalent to $\beta = 0.9$) and $\lambda_t = t$. SPS* (Garrigos et al., 2023) and IMA-SPS (Gower et al., 2025) require the exact optimal values $f_i(x^*)$. To approximate these quantities we run deterministic (full-batch) subgradient descent for $50,000$ iterations and treat the final iterate as $x^*$. All methods are trained for 100 epochs with batch size $n/10 = 30$. Every experiment is repeated three times with independent data draws; mean trajectories and one-standard-deviation bands are plotted in Figures 2c and 2d.

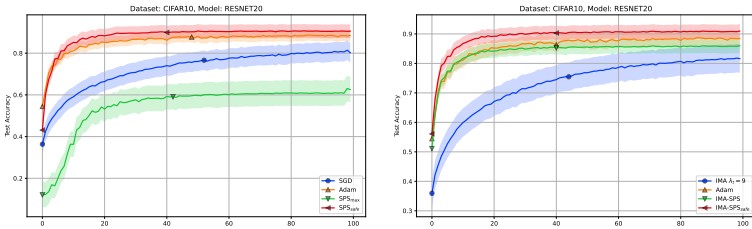

Figure 3: Test accuracy of ResNet20 on CIFAR-10. **Left:** SSM-based methods. **Right:** IMA-based methods.

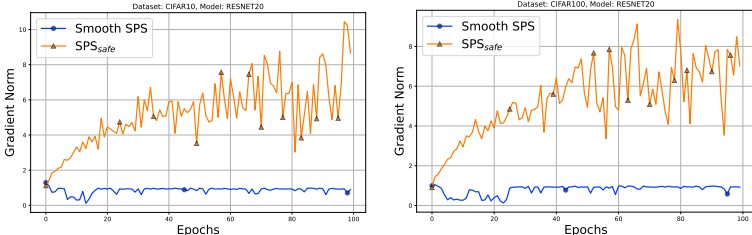

Figure 4: Gradient Norms during training of ResNet20. **Left:** Trained on CIFAR-10. **Right:** Trained on CIFAR-100.

The safeguarded Polyak rules dominate the competition on this non-smooth problem. In the plain-SSM setting (left panel) the best tuned $SPS_{safe}$ consistently outperforms both tuned fixed-step size SSM and $SPS^*$, achieving lower final error and faster early progress. The IMA experiments (right panel) paint the same picture: IMA-$SPS_{safe}$ is superior to IMA-SPS, and the momentum coefficient $\lambda_t = 9$ is clearly preferable to $\lambda_t = t$. These results confirm that the safeguard delivers practical gains in addition to its theoretical advantages.

### 4.2 APPLICATIONS ON DNNs

**Comparison with other optimizers.** We assess the safeguarded step sizes on image-classification benchmarks. ResNet-20/32 (He et al., 2016) models are trained on CIFAR-10/100 (Krizhevsky et al., 2009) with standard augmentation (random crop, horizontal flip, channel-wise normalisation (DeVries, 2017)). All runs are executed on NVIDIA RTX 6000 Ada GPUs for 100 epochs.[2] Baselines include tuned SSM, tuned IMA with $\lambda_t = 9$, Adam (Kingma & Ba, 2015), and $SPS_{max}$ and IMA-SPS. We compare these against the proposed safeguarded rules $SPS_{safe}$ and IMA-$SPS_{safe}$. For more details and more experiments we refer to the appendix. In Figure 3, we observe that in both SSM-based and IMA-based methods our proposed safeguarded step size have superior generalization performance.

**Comparison of Gradient Norms.** In Figure 4, we track the subgradient magnitude $\|g_i^t\|$ at the end of each epoch when training with Smooth $SPS_{max}$ versus $SPS_{safe}$ for ResNet-20 in CIFAR-10/100. Empirically, Smooth $SPS_{max}$ drives (sub)gradients to very small values, whereas $SPS_{safe}$ maintains noticeably larger norms. This behaviour is desirable: the safeguarded Polyak rule in $SPS_{safe}$ not only prevents division by vanishing gradients in its denominator, but also mitigates gradient collapse by preventing the gradient norms themselves from approaching zero.

## 5 CONCLUSION

In this work, we introduced *safeguarded* Polyak step sizes providing convergence rate $\mathcal{O}(T^{-1/2})$ to a neighborhood of the solution for stochastic, convex–Lipschitz objectives, the first Polyak rule to do so without assuming interpolation or the knowledge of $f_i(x^*)$. We extend the same idea to IMA, providing convergence for both the Cesàro average and the *last* iterate. Future work could explore techniques that eliminate the constant safeguard $M$ while retaining the current rates thus making the method parameter-free. Another promising direction is to extend the analysis to structured non-convex settings, such as weakly convex or PL objectives.

---

[2]All optimizers in the deep-learning experiments are run for the same fixed number of epochs with identical data augmentation, batch size, and weight decay, so the stopping criterion is consistent across methods.

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

# Supplementary Material

The Supplementary Material is organized as follows: Appendix A presents the proofs of the theoretical guarantees from the main paper. In Appendix B, we provide additional experiments.

## A    PROOFS

In this section, we present the proofs of the main theoretical results presented in the main paper, i.e. Proposition 2.1 and the convergence guarantees of $\text{SPS}_{safe}$ and $\text{IMA-SPS}_{safe}$. We restate the main theorems here for completeness.

### A.1    PROOF OF PROPOSITION 2.1

**Proposition.** *SSM with $\text{SPS}_{safe}$ and $M = c^2$ is algebraically equivalent to Clipped SSM with the adaptive step size $\tilde{\gamma}_t = \frac{f_i(x^t) - \ell_i^*}{c \max\{c, \|g_i^t\|\}}$.*

*Proof.* We have

$$x^{t+1} = x^t - \gamma_t g_i^t = x^t - \frac{f_i(x^t) - \ell_i^*}{\max\{c^2, \|g_i^t\|^2\}} g_i^t$$

$$= x^t - \frac{f_i(x^t) - \ell_i^*}{\min\left\{1, \frac{c}{\|g_i^t\|}\right\} \max\{c^2, \|g_i^t\|^2\}} \min\left\{1, \frac{c}{\|g_t\|}\right\} g_i^t$$

$$= x^t - \frac{f_i(x^t) - \ell_i^*}{\min\left\{1, \frac{c}{\|g_i^t\|}\right\} \max\{c^2, \|g_i^t\|^2\}} \text{clip}_c\left(g_i^t\right)$$

$$= x^t - \frac{f_i(x^t) - \ell_i^*}{c \max\{c, \|g_i^t\|\}} \text{clip}_c\left(g_i^t\right),$$

where the last equality follows by discriminating cases:

- If $\|g_i^t\| \le c$, then:

$$\min\left\{1, \frac{c}{\|g_i^t\|}\right\} \max\{c^2, \|g_i^t\|^2\} = 1 \cdot c^2 = c^2 \qquad \text{and}$$
$$c \max\{c, \|g_i^t\|\} = c\|g_i^t\|.$$

- If $\|g_i^t\| > c$, then:

$$\min\left\{1, \frac{c}{\|g_i^t\|}\right\} \max\{c^2, \|g_i^t\|^2\} = \frac{c}{\|g_i^t\|} \cdot \|g_i^t\|^2 = c\|g_i^t\| \qquad \text{and}$$
$$c \max\{c, \|g_i^t\|\} = c^2.$$

This completes the proof.                                                                 □

### A.2    PROOF OF THEOREM 3.1

**Theorem.** *Consider the iterates of SSM with the step size ($\text{SPS}_{safe}$). Then*

$$\mathbb{E}[f(\overline{x}^T) - f(x^*)] \le \frac{\sqrt{\max\{G^2, M\}}\|x^0 - x^*\|}{\sqrt{T}} + \sqrt{\frac{\max\{G^2, M\}}{M}}\sigma^2,$$

*where $\overline{x}^T = \frac{1}{T} \sum_{t=0}^{T-1} x^t$.*

*Proof of Theorem 3.1.* We have

$$\|x^{t+1} - x^*\|^2 - \|x^t - x^*\|^2$$

$$= -2\gamma_t \langle g_i^t, x^t - x^* \rangle + \gamma_t^2 \|g_i^t\|^2$$

$$\overset{\text{convexity}}{\leq} -2\gamma_t[f_i(x^t) - f_i(x^*)] + \gamma_t^2 \|g_i^t\|^2$$

$$= -\frac{2[f_i(x^t) - \ell_i^*][f_i(x^t) - f_i(x^*)]}{\max\{\|g_i^t\|^2, M\}} + \frac{[f_i(x^t) - \ell_i^*]^2}{(\max\{\|g_i^t\|^2, M\})^2}\|g_i^t\|^2$$

$$= -\frac{2[f_i(x^t) - \ell_i^*][f_i(x^t) - f_i(x^*)]}{\max\{\|g_i^t\|^2, M\}} + \frac{[f_i(x^t) - \ell_i^*]^2}{\max\{\|g_i^t\|^2, M\}} \frac{\|g_i^t\|^2}{\max\{\|g_i^t\|^2, M\}}$$

$$\leq -\frac{2[f_i(x^t) - \ell_i^*][f_i(x^t) - f_i(x^*)]}{\max\{\|g_i^t\|^2, M\}} + \frac{[f_i(x^t) - \ell_i^*]^2}{\max\{\|g_i^t\|^2, M\}}$$

$$= \frac{-2[f_i(x^t) - \ell_i^*][f_i(x^t) - f_i(x^*)] + [f_i(x^t) - \ell_i^*]^2}{\max\{\|g_i^t\|^2, M\}}$$

$$= \frac{(f_i(x^*) - \ell_i^*)^2 - (f_i(x^t) - f_i(x^*))^2}{\max\{\|g_i^t\|^2, M\}}$$

$$= -\frac{(f_i(x^t) - f_i(x^*))^2}{\max\{\|g_i^t\|^2, M\}} + \frac{(f_i(x^*) - \ell_i^*)^2}{\max\{\|g_i^t\|^2, M\}}$$

$$\leq -\frac{(f_i(x^t) - f_i(x^*))^2}{\max\{G^2, M\}} + \frac{(f_i(x^*) - \ell_i^*)^2}{M},$$

because $\max\{\|g_i^t\|^2, M\} \geq M$, $\|g_i^t\| \leq G$ and the function $z \mapsto \max\{M, z\}$ is increasing. Taking expectation conditional on $x^t$ we get

$$\mathbb{E}[\|x^{t+1} - x^*\|^2 | x^t] - \|x^t - x^*\|^2 \leq -\frac{\mathbb{E}[f_i(x^t) - f_i(x^*)]^2}{\max\{G^2, M\}} + \frac{\sigma^4}{M}$$

$$\leq -\frac{(\mathbb{E}[f_i(x^t) - f_i(x^*)])^2}{\max\{G^2, M\}} + \frac{\sigma^4}{M}$$

$$= -\frac{[f(x^t) - f(x^*)]^2}{\max\{G^2, M\}} + \frac{\sigma^4}{M},$$

where the second inequality follows by Jensen's inequality. Taking expectation again and using the tower property we have

$$\mathbb{E}\|x^{t+1} - x^*\|^2 \leq \mathbb{E}\|x^t - x^*\|^2 - \frac{\mathbb{E}[f(x^t) - f(x^*)]^2}{\max\{G^2, M\}} + \frac{\sigma^4}{M}.$$

Now summing up for $t = 0, \ldots, T - 1$ and telescoping we have

$$\frac{1}{\max\{G^2, M\}} \sum_{t=0}^{T-1} \mathbb{E}[f(x^t) - f(x^*)]^2 \leq \|x^0 - x^*\|^2 - \mathbb{E}\|x^T - x^*\|^2 + T\frac{\sigma^4}{M}$$

$$\leq \|x^0 - x^*\|^2 + T\frac{\sigma^4}{M}.$$

Now by Jensen's inequality (twice) we get

$$\mathbb{E}[f(\overline{x}^T) - f(x^*)] \leq \frac{1}{T} \sum_{t=0}^{T-1} \mathbb{E}[f(x^t) - f(x^*)]$$

$$\leq \sqrt{\frac{1}{T} \sum_{t=0}^{T-1} \mathbb{E}[f(x^t) - f(x^*)]^2}$$

$$\leq \sqrt{\frac{\max\{G^2, M\}\|x^0 - x^*\|^2}{T} + \frac{\max\{G^2, M\}\sigma^4}{M}}$$

$$\leq \frac{\sqrt{\max\{G^2, M\}}\|x^0 - x^*\|^2}{\sqrt{T}} + \sqrt{\frac{\max\{G^2, M\}}{M}}\sigma^2,$$

because $\sqrt{x + y} \leq \sqrt{x} + \sqrt{y}$. This completes the proof. $\qquad\square$

## A.3 PROOFS OF THEOREMS 3.4 AND 3.5

The next unified statement subsumes our two main momentum theorems, and we prove them using similar steps.

**Theorem.** *Consider the iterates of IMA with the step size (IMA-SPS$_{safe}$).*

- *If $\lambda_t = t$, then:*

$$\mathbb{E}[f(x^{T-1}) - f(x^*)] + \frac{1}{T}\sum_{t=0}^{T-1} t\,\mathbb{E}[B_f(x^{t-1}, x^t)] \le \frac{\sqrt{\max\{G^2, M\}}\|x^0 - x^*\|}{\sqrt{T}} + \sqrt{\frac{\max\{G^2, M\}}{M}}\sigma^2.$$

- *If $(\lambda_t)_{t>0}$ is a decreasing sequence of nonnegative reals, then:*

$$\mathbb{E}[f(\overline{x}^T) - f(x^*)] + \sum_{t=0}^{T-1} \frac{\lambda_t}{T}\,\mathbb{E}[B_f(x^{t-1}, x^t)] \le \frac{\sqrt{\max\{G^2, M\}}\|x^0 - x^*\|}{\sqrt{T}} + \sqrt{\frac{\max\{G^2, M\}}{M}}\sigma^2,$$

*where $\overline{x}^T = \frac{1}{T}\sum_{t=0}^{T-1} x^t$.*

*Proofs of Theorems 3.4 and 3.5.* We have

$$\|z^{t+1} - x^*\|^2 - \|z^t - x^*\|^2$$

$$= -2\eta_t\langle g_i^t, z^t - z^*\rangle + \eta_t^2\|g_i^t\|^2$$

$$= -2\eta_t\langle g_i^t, x^t - x^*\rangle - 2\eta_t\lambda_t\langle g_i^t, x^t - x^{t-1}\rangle + \eta_t^2\|g_i^t\|^2$$

$$\overset{\text{convexity}}{\le} -2\eta_t[f_i(x^t) - f_i(x^*) + \lambda_t\langle g_i^t, x^t - x^{t-1}\rangle] + \eta_t^2\|g_i^t\|^2$$

$$= -\frac{2[f_i(x^t) - \ell_i^* + \lambda_t\langle g_i^t, x^t - x^{t-1}\rangle]_+[f_i(x^t) - f_i(x^*) + \lambda_t\langle g_i^t, x^t - x^{t-1}\rangle]}{\max\{\|g_i^t\|^2, M\}}$$

$$+ \frac{[f_i(x^t) - \ell_i^* + \lambda_t\langle g_i^t, x^t - x^{t-1}\rangle]_+^2}{(\max\{\|g_i^t\|^2, M\})^2}\|g_i^t\|^2.$$

Now for easier notation we set $q = f_i(x^t) + \lambda_t\langle g_i^t, x^t - x^{t-1}\rangle$, thus we continue:

$$\|z^{t+1} - x^*\|^2 - \|z^t - x^*\|^2$$

$$= \frac{-2(q - \ell_i^*)_+ \cdot (q - f_i(x^*))}{\max\{\|g_i^t\|^2, M\}} + \frac{(q - \ell_i^*)_+^2}{\max\{\|g_i^t\|^2, M\}} \cdot \frac{\|g_i^t\|^2}{\max\{\|g_i^t\|^2, M\}}$$

$$\le \frac{-2(q - \ell_i^*)_+ \cdot (q - f_i(x^*))}{\max\{\|g_i^t\|^2, M\}} + \frac{(q - \ell_i^*)_+^2}{\max\{\|g_i^t\|^2, M\}}$$

$$= \frac{-2(q - \ell_i^*)_+ \cdot (q - f_i(x^*)) + (q - \ell_i^*)_+^2}{\max\{\|g_i^t\|^2, M\}}$$

$$\overset{(\star)}{\le} \frac{(f_i(x^*) - \ell_i^*)^2 - (q - f_i(x^*))_+^2}{\max\{\|g_i^t\|^2, M\}}$$

$$= -\frac{(f_i(x^t) - f_i(x^*) + \lambda_t\langle g_i^t, x^t - x^{t-1}\rangle)_+^2}{\max\{\|g_i^t\|^2, M\}} + \frac{[f_i(x^*) - \ell_i^*]^2}{\max\{\|g_i^t\|^2, M\}}$$

$$\overset{(\star\star)}{\le} -\frac{[f_i(x^t) - f_i(x^*) + \lambda_t\langle g_i^t, x^t - x^{t-1}\rangle]_+^2}{\max\{G^2, M\}} + \frac{[f_i(x^*) - \ell_i^*]^2}{M}.$$

Let's explain inequality $(\star)$, which is:

$$-2(q - \ell_i^*)_+ \cdot (q - f_i(x^*)) + (q - \ell_i^*)_+^2 \le (f_i(x^*) - \ell_i^*)^2 - (q - f_i(x^*))_+^2 \qquad (\star)$$

Note that $\ell_i^* \le f_i(x^*)$ so $q - \ell_i^* \ge q - f_i(x^*)$. Hence if $q - \ell_i^* \le 0$ inequality $(\star)$ reduces to the obvious $0 \le [f_i(x^t) - \ell_i^*]^2$. Now assume that $q - \ell_i^* > 0$. Then

$$-2(q - f_i^*)_+ \cdot (q - f_i(x^*)) + (q - f_i^*)_+^2 = -2(q - f_i^*)(q - f_i(x^*)) + (q - f_i^*)^2$$

$$= (q - f_i^* - (q - f_i(x^*)))^2 - (q - f_i(x^*))^2$$

$$= (f_i(x^*) - f_i^*)^2 - (q - f_i(x^*))^2$$

$$\le (f_i(x^*) - f_i^*)^2 - (q - f_i(x^*))_+^2,$$

as wanted. Now, inequality $(\star\star)$ follows from $\max\{\|g_i^t\|^2, M\} \geq M$ and $\max\{\|g_i^t\|^2, M\} \leq \max\{G^2, M\}$ because $f_i$ is $G$-Lipschitz.

Now taking expectation and using Lemmas A.1 and A.2 we get

$$\mathbb{E}\|z^{t+1} - x^*\|^2 \leq \mathbb{E}\|z^t - x^*\|^2 - \frac{\mathbb{E}[f(x^t) - f(x^*) + \lambda_t\langle\partial f(x^t), x^t - x^{t-1}\rangle]_+^2}{\max\{G^2, M\}} + \frac{\sigma^4}{M}$$

$$= \mathbb{E}\|z^t - x^*\|^2 - \frac{\mathbb{E}[(1 + \lambda_t)[f(x^t) - f(x^*)] - \lambda_t[f(x^{t-1}) - f(x^*)] + \lambda_t B_f(x^{t-1}, x^t)]_+^2}{\max\{G^2, M\}} + \frac{\sigma^4}{M},$$

so

$$\mathbb{E}[(1 + \lambda_t)[f(x^t) - f(x^*)] - \lambda_t[f(x^{t-1}) - f(x^*)] + \lambda_t B_f(x^{t-1}, x^t)]_+^2$$

$$\leq \max\{G^2, M\}\,\mathbb{E}\|z^t - x^*\|^2 - \max\{G^2, M\}\,\mathbb{E}\|z^{t+1} - x^*\|^2 + \frac{\max\{G^2, M\}}{M}\sigma^4.$$

Now let $\Delta_t = (1 + \lambda_t)[f(x^t) - f(x^*)] - \lambda_t[f(x^*) - f(x^*)] + \lambda_t B_f(x^{t-1}, x^t)$, sum for $t = 0, \dots, T-1$ and use Jensen to get

$$\frac{\max\{G^2, M\}\|x^0 - x^*\|^2}{T} + \frac{\max\{G^2, M\}}{M}\sigma^4$$

$$\geq \frac{1}{T}\sum_{t=0}^{T-1}\mathbb{E}[\Delta_t]_+^2$$

$$\geq \left(\frac{1}{T}\sum_{t=0}^{T-1}\mathbb{E}[\Delta_t]\right)_+^2,$$

which means that

$$\left(\frac{1}{T}\sum_{t=0}^{T-1}\mathbb{E}[\Delta_t]\right)_+ \leq \sqrt{\frac{G^2\|x^0 - x^*\|^2}{T} + \frac{\max\{G^2, M\}}{M}\sigma^4}$$

$$\leq \frac{\sqrt{\max\{G^2, M\}}\|x^0 - x^*\|}{\sqrt{T}} + \sqrt{\frac{\max\{G^2, M\}}{M}}\sigma^2.$$

Now

$$\sum_{t=0}^{T-1}\mathbb{E}[\Delta_t] = \sum_{t=0}^{T-1}(1 + \lambda_t)\,\mathbb{E}[f(x^t) - f(x^*)] - \lambda_t\,\mathbb{E}[f(x^{t-1}) - f(x^*)] + \lambda_t\,\mathbb{E}[B_f(x^{t-1}, x^t)]$$

$$= \sum_{t=0}^{T-1}\lambda_t\,\mathbb{E}[B_f(x^{t-1}, x^t)] + \sum_{t=0}^{T-1}\mathbb{E}[f(x^t) - f(x^*)] + \sum_{t=0}^{T-2}(\lambda_t - \lambda_{t+1})\,\mathbb{E}[f(x^t) - f(x^*)]$$

$$+ \lambda_{T-1}\,\mathbb{E}[f(x^{T-1}) - f(x^*)].$$

Now if $\lambda_t = t$ then

$$\sum_{t=0}^{T-1}\lambda_t\,\mathbb{E}[B_f(x^{t-1}, x^t)] + \sum_{t=0}^{T-1}\mathbb{E}[f(x^t) - f(x^*)] + \sum_{t=0}^{T-2}(\lambda_t - \lambda_{t+1})\,\mathbb{E}[f(x^t) - f(x^*)]$$

$$+ \lambda_{T-1}\,\mathbb{E}[f(x^{T-1}) - f(x^*)]$$

$$= \sum_{t=0}^{T-1}t\,\mathbb{E}[B_f(x^{t-1}, x^t)] + T \cdot \mathbb{E}[f(x^{T-1}) - f(x^*)] \geq 0,$$

so we get

$$\mathbb{E}[f(x^{T-1}) - f(x^*)] + \sum_{t=0}^{T-1}\frac{t}{T}\,\mathbb{E}[B_f(x^{t-1}, x^t)] \leq \frac{\sqrt{\max\{G^2, M\}}\|x^0 - x^*\|}{\sqrt{T}} + \sqrt{\frac{\max\{G^2, M\}}{M}}\sigma.$$

This completes the proof of Theorem 3.5.

If $(\lambda_t)_{t>0}$ is decreasing then

$$\sum_{t=0}^{T-1} \lambda_t \, \mathbb{E}[B_f(x^{t-1}, x^t)] + \sum_{t=0}^{T-1} \mathbb{E}[f(x^t) - f(x^*)] + \sum_{t=0}^{T-2} (\lambda_t - \lambda_{t+1}) \, \mathbb{E}[f(x^t) - f(x^*)]$$

$$+ \lambda_{T-1} \, \mathbb{E}[f(x^{T-1}) - f(x^*)]$$

$$\geq \sum_{t=0}^{T-1} \lambda_t \, \mathbb{E}[B_f(x^{t-1}, x^t)] + \sum_{t=0}^{T-1} \mathbb{E}[f(x^t) - f(x^*)]$$

$$\geq \sum_{t=0}^{T-1} \lambda_t \, \mathbb{E}[B_f(x^{t-1}, x^t)] + T \cdot \mathbb{E}[f(\overline{x}^T) - f(x^*)],$$

so we get

$$\mathbb{E}[f(\overline{x}^T) - f(x^*)] + \sum_{t=0}^{T-1} \lambda_t \, \mathbb{E}[B_f(x^{t-1}, x^t)] \leq \frac{\sqrt{\max\{G^2, M\}}\|x^0 - x^*\|}{\sqrt{T}} + \sqrt{\frac{\max\{G^2, M\}}{M}} \sigma.$$

This completes the proof of Theorem 3.4. $\qquad\qquad\square$

Here we provide the two auxiliary lemmas used in the previous proof.

**Lemma A.1** ((Gower et al., 2025): Lem. B.3). *For any random variable $X$ and positive-valued random variable $Y$, it holds*

$$\mathbb{E}\left[\frac{(X)_+^2}{Y}\right] \geq \frac{(\mathbb{E}\,X)_+^2}{\mathbb{E}\,Y}.$$

**Lemma A.2** ((Gower et al., 2025): Lem. C.3). *For any $x^t, x^t, x^* \in \mathbb{R}^d$ and $\lambda_t \geq 0$ it holds*

$$f(x^t) - f(x^*) + \lambda_t \langle \partial f(x^t), x^t - x^{t-1} \rangle$$
$$= (1 + \lambda_t)[f(x^t) - f(x^*)] - \lambda_t[f(x^{t-1}) - f(x^*)] + \lambda_t B_f(x^{t-1}, x^t),$$

*where $B_f(x, y) = f(x) - f(y) - \langle \partial f(y), x - y \rangle$ is the Bregman divergence.*

# B  MORE DEEP LEARNING EXPERIMENTS AND PARAMETER SETTINGS

In this section, we list the parameters, architectures and hardware that we used for the deep learning experiments. The information is collected in Table 2. We also include some extra experiments (ResNet20 in CIFAR-100 and ResNet32 in CIFAR-10/100) in Figures 5 to 7.

| Hyper-parameter | Value |
| --- | --- |
| Datasets | CIFAR-10/100 (Krizhevsky et al., 2009) |
| Architecture | ResNet 20/32 (He et al., 2016) |
| GPUs | 1x Nvidia RTX 6000 Ada Generation |
| Batch-size | 128 |
| Epochs | 100 |
| Weight Decay | 0.0 |

Table 2: Experimental details

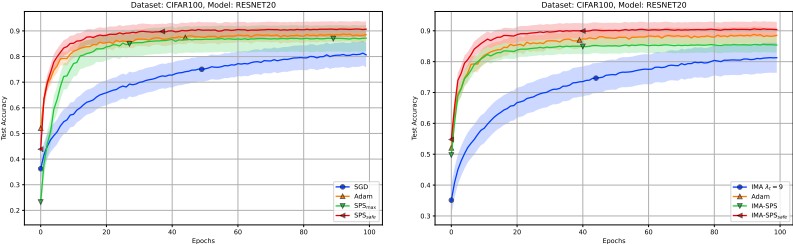

Figure 5: Test accuracy of ResNet20 on CIFAR-100. **Left:** SSM-based methods. **Right:** IMA-based methods.

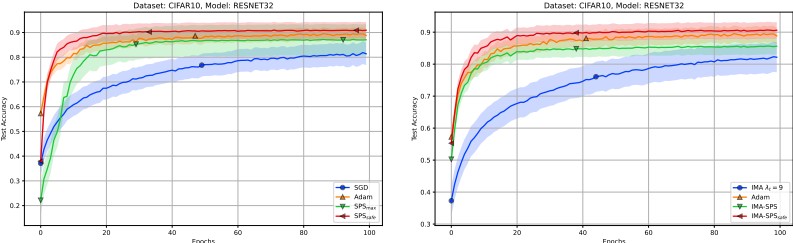

Figure 6: Test accuracy of ResNet32 on CIFAR-10. **Left:** SSM-based methods. **Right:** IMA-based methods.

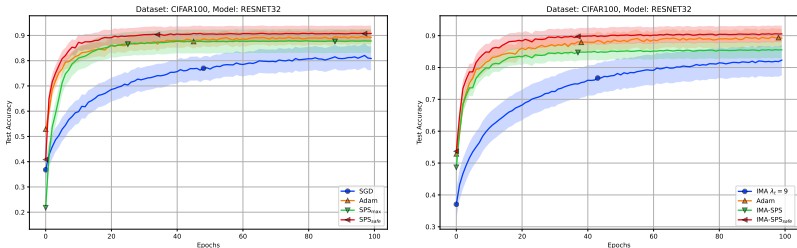

Figure 7: Test accuracy of ResNet32 on CIFAR-100. **Left:** SSM-based methods. **Right:** IMA-based methods.

## B.1 COMPARISON OF THE GRADIENT NORM

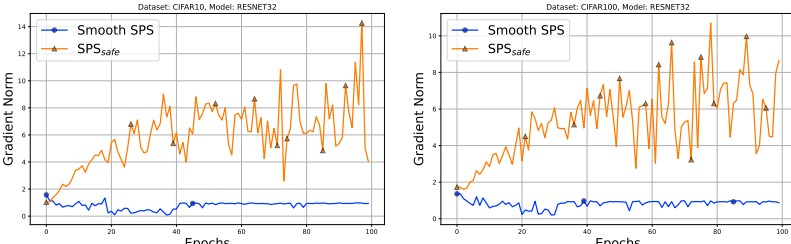

Figure 8: Gradient Norms during training of ResNet20. **Left:** Trained on CIFAR-10. **Right:** Trained on CIFAR-100.

## C EXTRA SENSITIVITY ANALYSIS

In this appendix we complement the main sensitivity study for the safeguard $M$ by providing additional experiments on both convex and deep-learning benchmarks. We systematically vary $M$ over a wide range and report the generalization performance vs the value of $M$ for SSM and IMA variants. These plots illustrate that choosing $M = 1.0$ works well in practice.

### C.1 CONVEX

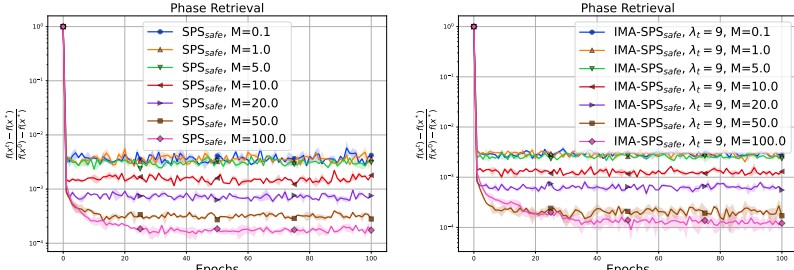

Figure 9: Sensitivity Analysis for various safeguards $M$ for Phase Retrieval. **Left:** SSM. **Right:** IMA

### C.2 DEEP LEARNING

#### C.2.1 SSM

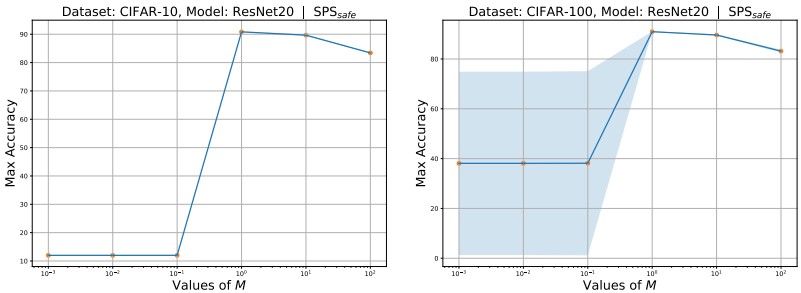

Figure 10: Sensitivity Analysis for various safeguards $M$ for ResNet20. **Left:** Trained on CIFAR-10. **Right:** Trained on CIFAR-100.

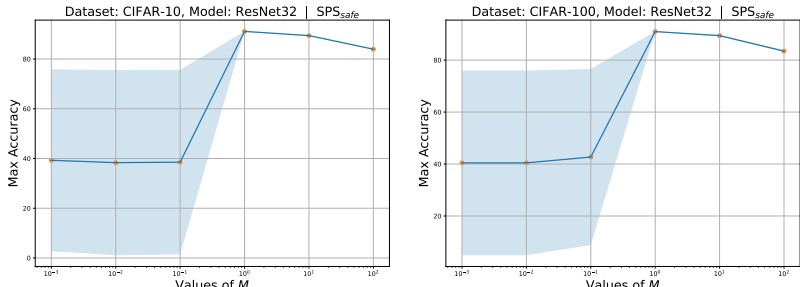

Figure 11: Sensitivity Analysis for various safeguards $M$ for ResNet32. **Left:** Trained on CIFAR-10. **Right:** Trained on CIFAR-100.

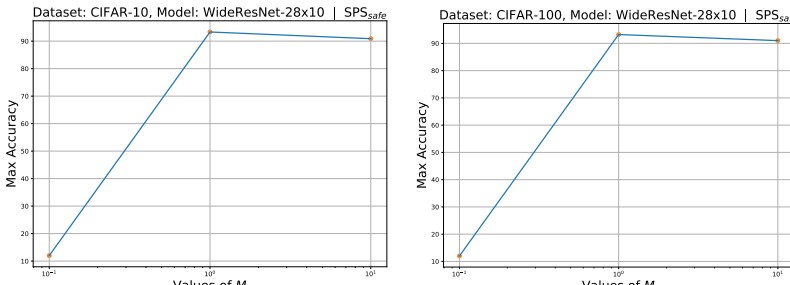

Figure 12: Sensitivity Analysis for various safeguards $M$ for WideResNet 28x10. **Left:** Trained on CIFAR-10. **Right:** Trained on CIFAR-100.

### C.2.2 IMA

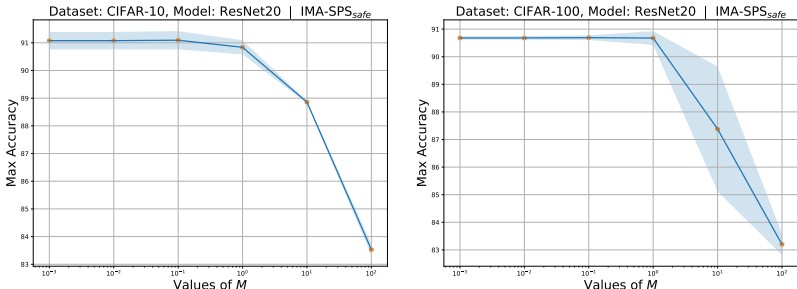

Figure 13: Sensitivity Analysis for various safeguards $M$ for ResNet20. **Left:** Trained on CIFAR-10. **Right:** Trained on CIFAR-100.

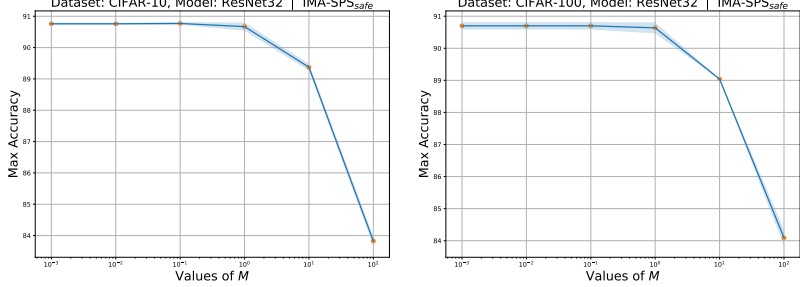

Figure 14: Sensitivity Analysis for various safeguards $M$ for ResNet32. **Left:** Trained on CIFAR-10. **Right:** Trained on CIFAR-100.

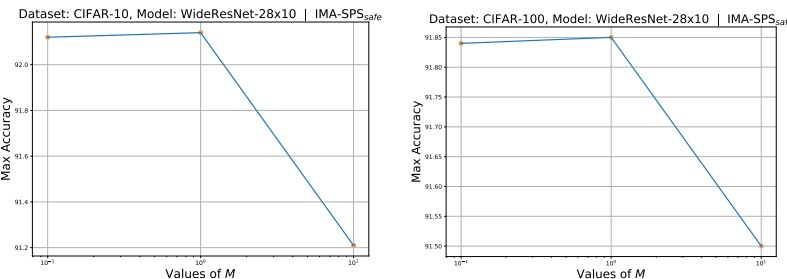

Figure 15: Sensitivity Analysis for various safeguards $M$ for WideResNet 28x10. **Left:** Trained on CIFAR-10. **Right:** Trained on CIFAR-100.

# D    SMOOTHING TRICK FOR $M$

Motivated by the sensitivity analyses above, in this appendix, we investigate a simple smoothing strategy that replaces the fixed safeguard $M$ with an exponential moving average $M_t$ of past squared gradients. This adaptive rule is designed to reduce manual tuning while preserving the stabilizing effect of the safeguard. We present the corresponding update, provide a practical recommendation for the smoothing parameter $\beta$, and compare the resulting $M_t$ against the best-tuned fixed $M$ on CIFAR-10/100 and several architectures.

The $\text{SPS}_{safe}$ takes the following form:

$$\gamma_t = \frac{f_i(x^t) - \ell_i^*}{\max\{\|g_i^t\|^2, M_t\}}$$
$$M_t = \beta M_{t-1} + (1 - \beta)\|g_i^t\|^2,$$

with $M_0 = \|g_i^0\|^2$. For a good practical performance we recommend $\beta = 0.9$.

Table 3: Comparison of test accuracy of tuned $M$ vs Smooth $M_t$ for various model on CIFAR10.

| Model | Best $M$ | Smooth $M_t$ ($\beta = 0.9$) |
|---|---|---|
| ResNet20 | $90.84_{\pm 0.17}$ | $\mathbf{90.97}_{\pm 0.14}$ |
| ResNet32 | $90.80_{\pm 0.04}$ | $\mathbf{90.94}_{\pm 0.13}$ |
| WideResNet-28x10 | $\mathbf{93.03}$ | $92.99$ |

Table 4: Comparison of test accuracy of tuned $M$ vs Smooth $M_t$ for various model on CIFAR100.

| Model | Best $M$ | Smooth $M_t$ ($\beta = 0.9$) |
|---|---|---|
| ResNet20 | $90.86_{\pm 0.12}$ | $\mathbf{90.93}_{\pm 0.16}$ |
| ResNet32 | $90.97_{\pm 0.11}$ | $\mathbf{91.11}_{\pm 0.28}$ |
| WideResNet-28x10 | $\mathbf{93.24}$ | $93.22$ |

