# OpenReview forum: "Safeguarded Stochastic Polyak Step-Sizes for Non-smooth Optimization: Robust Performance Without Small (Sub)Gradients"
_ICLR.cc/2026/Conference — Submitted to ICLR 2026_

### Official Review · Reviewer_d8h5 · 2025-10-27

**Soundness:** 3
**Presentation:** 3
**Contribution:** 2
**Rating:** 4
**Confidence:** 4

**Summary:**

This paper introduces a novel Polyak-type step size, namely Safeguarded SPS. It modifies the function minimum value in the previous Polyak-type steps to a lower bound \(\ell_i^*\), eliminating the need for the interpolation assumption while avoiding reliance on oracle information. Additionally, it clips the gradient norm information to ensure the smoothness of the step size, which can effectively address the vanishing gradient problem in neural network training.

**Strengths:**

1. It provides a comprehensive review of the existing methods with Polyak-type step sizes analyzed in the convex non-smooth stochastic setting. A table is used to compare these methods, including SPS, SPS*, IMA, and the two methods proposed in this paper. Notably, the proposed methods can maintain a favorable convergence rate without requiring the interpolation condition or prior knowledge of \(f_i(x^*)\).

2. In the deep neural network (DNN) training section, key experimental data (especially the proportion of iterations using the Polyak step) and figures are presented to compare the proposed methods with \(SPS_{max}\) and \(Smooth SPS_{max}\). These data and figures are clearly explained, intuitively demonstrating the advantage of the proposed methods: their step sizes remain adaptive throughout the iteration process and never degenerate into constant values.

**Weaknesses:**

1. The experimental section only conducts experiments on Support Vector Machines (SVM), Phase Retrieval, and Deep Neural Networks (DNNs). More cases could be tested to verify whether favorable convergence results can be achieved. In the DNN training experiments, only ResNet-20/32 models and the CIFAR-10/100 datasets are used; experiments on different models and more complex datasets could be attempted. Moreover, the reported test accuracy (<90%) on CIFAR-10 appears lower than typical results for ResNet-20 with Adam (around 91–92%), indicating that further hyperparameter tuning or training optimization might be needed.

2. There seems to be no robust strategy for selecting \(\ell_i^*\) and \(M\). For tasks like neural network training, the \(\ell_i^*\) of the loss function may be easy to obtain, but are there any selection methods for other types of problems? The selection of \(\ell_i^*\) affects convergence results: if it is chosen too small, \(\sigma\) will become too large, thereby degrading convergence performance. Moreover, \(\ell_i^*\) may not be easily selectable in more complex problems. Additionally, in the sensitivity analysis of \(M\), experiments are only conducted for \(M = \{1, 10, 100\}\).

3. The proofs of the theorems are mostly standard, and the convexity assumption is used in the proofs.

4. There is a typo in the definition of \(f_i^*\) in Line 108: \(x\) is incorrectly written as \(x^*\).

**Questions:**

1. In the theorem proof section, the proofs are only provided for convex problems. However, the loss functions in practical neural network training are often highly non-convex. What difficulties are there in extending the convergence conclusions to non-convex problems?

2. Poor selection of \(\ell_i^*\) and \(M\) may lead to very poor convergence results, and they may be difficult to select in complex problems. How can this issue be addressed?

---

> ### Author Response · Authors · 2025-11-20
> **Author's Rebuttal**
>
> We thank the reviewer for their comments.
>
> **On experiments.**
>
> Our experimental design was intentional rather than minimal: we focused on (i) convex benchmarks (SVM, phase retrieval) where the theory applies exactly, and (ii) standard vision benchmarks (CIFAR-10/100 with ResNet-20/32) that are widely used to evaluate new optimizers. Following your suggestion, we have expanded the empirical section with preliminary results on WideResNet-28x10 for CIFAR-10/100, now included in Appendix C (a more exhaustive study will be carried out for the camera-ready, given current time constraints). Regarding your comment on Adam, our aim was to ensure a fair comparison under identical conditions across all methods, for this reason, we did not add additional “tricks” (e.g., weight decay) that are known to improve Adam’s test accuracy.
>
> **Regarding the selection of $\ell_i^\ast$ and $M$.**
>
> As discussed in the paper, $\ell_i^\ast$​ is simply a *lower bound* on the loss $f_i(x)$. In practice, most standard losses are nonnegative, so in the vast majority of applications one can safely set $\ell_i^\ast=0$. Naturally, the closer $\ell_i^\ast$ is to the true infimum $f_i^\ast=\inf f_i(x)$, the better the constants in the bounds, however, accurately identifying $f_i^\ast$​ is generally difficult in practice. In some special cases this is possible, see for example, Loizou et al., 2021, App. D.
>
> Regarding the choice of M, as you mentioned we already included a sensitivity study in the convex setting for M = {1.0, 10, 100.0}. In the revised version, we have expanded this analysis and added Appendix C: in the convex experiments we now sweep M = {0.1, 1.0, 5.0, 10.0, 20.0, 50.0, 100.0}, and in the deep-learning setting we report maximum accuracy as a function of M for M = {0.001, 0.01, 0.1, 1.0, 10.0, 100.0} across multiple datasets and architectures. These results consistently indicate that M=1.0 is a reliable default choice for practitioners. Finally, we added Appendix D, where we propose an exponentially averaged definition of M (see lines 1084–1088) that is tuning-free and works very well in practice.
>
>
> **On the convexity assumption.**
>
> We agree that our convergence analysis currently covers only convex Lipschitz objectives. This is a deliberate choice: tightly characterizing Polyak-type step sizes in the convex non-smooth regime is already technically challenging, as evidenced by the lack of such results in previous work. Indeed, the objective functions in most deep neural networks are both non-convex and non-smooth. In our case, we focus on the non-smooth setting and use convexity as our main tool in the proofs. Let us also highlight that there are recent works (see [Schaipp et al, 2025]) that highlight the agreement between convex non-smooth optimization theory and learning-rate scheduling for large model training. Following this paper, in our work, we treat the deep-learning experiments as empirical evidence that the convex insights transfer to non-convex practice (similar to the approach of [Schaipp et al, 2025]).
>
> The main obstacle to extending Polyak-type step sizes to the non-convex setting is that even their deterministic derivation fundamentally relies on convexity. In other words, Polyak steps are, by design, a convexity-based construction, and the usual arguments do not carry over in any straightforward way to non-convex objectives.
>
> Having said the above, in our opinion, having an analysis on convex and non-smooth regimes should not be considered a weakness of the work. Even if the proofs might look standard, this should not undermine the main contribution of our work: the proposal of the first practcal Polyak step-size methods for a convex non-smooth regime.
>
>
> **On typo:** Thanks for catching it! We have already fixed it in the updated document.
>
> **Reading your comments, we believe that all pointed weaknesses are simple clarifications (not reasons justifying a rejection score). If you agree that we managed to address all issues, please consider raising your mark to support our work. If you believe this is not the case, please let us know so that we have a chance to respond.**
>
>
>
> *References:*
>
> The Surprising Agreement Between Convex Optimization Theory and Learning-Rate Scheduling for Large Model Training, Schaipp F, Hägele A, Taylor A, Simsekli U, Bach F. ICML 2025.

---

### Official Review · Reviewer_Zq5v · 2025-10-30

**Soundness:** 3
**Presentation:** 3
**Contribution:** 3
**Rating:** 6
**Confidence:** 3

**Summary:**

This paper introduces Safeguarded Stochastic Polyak Step Size (SPSsafe) for stochastic subgradient methods designed to handle non-smooth convex optimization problems. Unlike prior SPS variants for non-smooth convex problems, SPSsafe does not require knowledge of the optimal solution or rely on interpolation assumptions. The paper establishes convergence guarantees and extends their approach by integrating momentum, maintaining the theoretical properties. The paper experiments on both convex optimization benchmarks and deep neural network training.

**Strengths:**

The paper is well motivated, building on the growing interest in SPS-type step-size rules, which have recently proven highly effective for stochastic optimization and deep learning. It makes a good contribution by extending this family of methods to the non-smooth setting. To the best of my understanding, this is the first SPS variant that offers provable convergence for non-smooth problems, or at least a practical one that does so without requiring knowledge of the optimal value.

There is a good comparison to the existing SPS step-size rules, in this way, I find the comparison good and useful to understand the contribution.

The work is also extensive, in the sense that it establishes convergence rate for both SGD and SGD with momentum.

The numerical results show clear benefits of the proposed approach.

**Weaknesses:**

The theoretical results could be better contrasted with results for general SGD. What is not clear, if there are some theoretical benefits of SPS-type step-size scheduling, compared to other step-size choices. I don’t think so, the O(1/sqrt(t)) convergence is standard for SGD for a problem with functions of this  structure. I don’t know if there are improvements for the scalar factor, or what. The contribution is still interesting, because there are practical benefits, but it would be good if this could be explained better.

The methodology in the experiments could be strengthened. The reported results are based on single runs, which makes it difficult to assess the statistical significance or robustness of the improvements. Including multiple runs with confidence intervals or standard deviations would give a clearer picture of performance variability and help substantiate the claimed advantages. More detailed analysis of sensitivity to hyperparameters and comparisons under consistent stopping criteria would also make the experimental evaluation more convincing.

**Questions:**

From my understanding, SPS step-sizes are particularly applicable because they are more robust to parameter tuning, e.g., like initial step-size. It would be nice to do some analysis on this compare to other algorithms.

---

> ### Author Response · Authors · 2025-11-20
> **Author's Rebuttal**
>
> We would like to thank the reviewer for their time and the positive evaluation. We appreciate the comments on the strengths of our work.
>
> **Comparison with SGD.**
>
> The main advantage of SPS_safe over SGD with constant or decreasing step size is that it removes the need to hand-tune the learning-rate, while still achieving the same $O(T^{-1/2})$ convergence rate in the convex, non-smooth setting. Classical SGD (and its variants) requires choosing step sizes based on problem-dependent quantities (e.g., Lipschitz constant, distance to the optimum, time horizon), and its performance is highly sensitive to these choices. In contrast, SPS_safe computes the step size adaptively from the current loss and (sub)gradient, using only a single safeguard parameter and no prior knowledge of problem constants.
>
> **On Experiments.**
>
> In the original version of our work, we already ran 3 runs and report the average performance (not a single run) for the convex experiments. In the updated version, per the reviewers' suggestion, we include the shaded region to indicate the general performance of the method (please see the updated PDF file) with confidence intervals. Moreover, regarding the sensitivity of M, we already had included a sensitivity study in the convex setting for M = {1.0, 10, 100.0}. In the revised version, we have expanded this analysis and it is added in Appendix C.
>
> Finally, regarding stopping criteria and fairness: All optimizers in the deep-learning experiments are run for the same fixed number of epochs (100) with identical data augmentation, batch size, and weight decay, so the stopping criterion is consistent across methods. We added a short sentence clarifying this to emphasise that comparisons are made under identical training budgets.
>
> **On robustness to parameter tuning.**
>
> Thanks for the suggestion. This is an interesting experimental setting. For this, let us note that our initial step-size is already always different, since it depends on the mini-batch $i\in[n]$. We run our algorithms in multiple seeds, so the 0-th mini-batch is different in each seed.
>
>
> **Thanks again for the review and the positive evaluation of our work. Reading your comments, we believe that all pointed weaknesses are simple clarifications. If you agree that we managed to address all issues, please consider raising your mark to clearly support our work. If you believe this is not the case, please let us know so that we have a chance to respond.**

---

### Official Review · Reviewer_6THh · 2025-10-31

**Soundness:** 3
**Presentation:** 3
**Contribution:** 3
**Rating:** 6
**Confidence:** 3

**Summary:**

The paper presents a new optimization technique called Safeguarded Stochastic Polyak Step-size ($SPS_{safe}$), which solves essential problems in non-smooth optimization for machine learning applications. The previous Polyak-type methods proved unusable because they needed data interpolation and unachievable "oracle" knowledge about the best solution. The main advancement of $SPS_{safe}$ introduces a basic protection system that stops the step-size from growing excessively when gradient values approach zero through the addition of a denominator floor value. The method provides the first Polyak-style optimization technique that proves convergence without needing unrealistic assumptions. The research develops a momentum-based algorithm from this concept and proves its connection to the popular gradient clipping heuristic. The proposed method demonstrates its effectiveness through multiple experiments on convex problems and deep neural networks, which show it achieves faster convergence and lower variance while outperforming standard adaptive optimizers by solving vanishing gradient problems.

**Strengths:**

1. A novel step-size for SSM without strong assumptions. The paper presents $SPS_{safe}$ as the first Polyak-type step-size for SSM, which proves convergence in stochastic convex non-smooth environments without needing interpolation conditions or oracle access to $f_{i}(x^{*})$. The algorithm reaches a solution neighborhood with a convergence rate of $\mathcal{O}(1/\sqrt{T})$.
2. The research conducts an extensive evaluation of the protection system's advantages. The paper demonstrates through formal algebraic methods that $SPS_{safe}$ operates identically to clipped SSM with an adaptive learning rate, thus establishing the first theoretical convergence proof for this method in stochastic environments.
3. Extension to momentum with rigorous guarantees. The safeguarding concept extends to the IMA framework, which results in $IMA-SPS_{safe}$. The method delivers the first adaptive momentum technique for non-smooth optimization, which proves convergence through rigorous analysis without needing oracle access, while supporting both Cesaro average and last iterate convergence.

**Weaknesses:**

1. The system performs image classification using ResNet-20/32 models on CIFAR data and synthetic convex tasks, but lacks training on large NLP datasets with transformers and multiple architecture types.
2. The paper establishes formal convergence results only for functions that are both convex and Lipschitz continuous. The method shows excellent performance on deep neural networks with non-convex structures, but lacks theoretical justification for this success.
3. The process of choosing $M$ value requires a delicate decision because increasing $M$ reduces the final error range, but it can decrease the speed of convergence at the beginning. The paper fails to study or offer methods to determine the optimal point for balancing this trade-off.

**Questions:**

See weakness.

---

> ### Author Response · Authors · 2025-11-20
> **Author's Rebuttal**
>
> We would like to thank the reviewer for their time and the positive evaluation. We appreciate the comments on the strengths of our work.
>
> **On the Scope of Experiments.**
>
> In the experiments of this work, we deliberately focused on (a) convex benchmarks, where our theory applies exactly, and (b) widely used vision benchmarks (CIFAR-10/100 with ResNet-20/32), which are standard for evaluating new optimization methods. Our algorithm is, however, architecture-agnostic and can be directly applied to Transformers and other NLP models. We do not have the computational capacity to run this kind of experiment in the rebuttal period (just one week), but we can aim to include small-scale NLP experiments in the camera-ready version. Having said that, in our opinion, the lack of NLP experiments should not be seen as a limitation of a theory-oriented paper.
>
> **On the convexity assumption.**
>
> We agree that our convergence analysis currently covers only convex Lipschitz objectives. This is a deliberate choice: tightly characterizing Polyak-type step sizes in the convex non-smooth regime is already technically challenging, as evidenced by the lack of such results in previous work. Indeed, the objective functions in most deep neural networks are both non-convex and non-smooth. In our case, we focus on the non-smooth setting and use convexity as our main tool in the proofs. Let us also highlight that there are recent works (see [Schaipp et al, 2025]) that highlight the agreement between convex non-smooth optimization theory and learning-rate scheduling for large model training. Following this paper, in our work, we treat the deep-learning experiments as empirical evidence that the convex insights transfer to non-convex practice (similar to the approach of [Schaipp et al, 2025]).
>
>
> **Choice of safeguard M.**
>
> We agree that the safeguard M induces a trade-off: a larger M tightens the final error neighborhood while potentially slowing early progress, whereas a smaller M accelerates initially but leads to a larger neighborhood. In the original submission, we already included a sensitivity study in the convex setting for M = {1.0, 10, 100.0}, showing that SPS_safe and IMA-SPS_safe are robust across this range. In the revised version, we have expanded this analysis and added Appendix C: in the convex experiments we now sweep M = {0.1, 1.0, 5.0, 10.0, 20.0, 50.0, 100.0}, and in the deep-learning setting we report maximum accuracy as a function of M for M = {0.001, 0.01, 0.1, 1.0, 10.0, 100.0} across multiple datasets and architectures. These results consistently indicate that M=1.0 is a reliable default choice for practitioners. Finally, we added Appendix D, where we propose an exponentially averaged definition of M (see lines 1089–1093) that is tuning-free and works very well in practice.
>
> Reference:
> The Surprising Agreement Between Convex Optimization Theory and Learning-Rate Scheduling for Large Model Training, Schaipp F, Hägele A, Taylor A, Simsekli U, Bach F. ICML 2025.
>
>
> **Thanks again for the review and the positive evaluation of our work.
> After reading your comments, we believe all the pointed weaknesses are simple clarifications. If you agree that we managed to address all issues, please consider raising your mark to clearly support our work. If you believe this is not the case, please let us know so that we have a chance to respond.**

---

### Official Review · Reviewer_o9nW · 2025-11-01

**Soundness:** 2
**Presentation:** 1
**Contribution:** 1
**Rating:** 2
**Confidence:** 4

**Summary:**

This paper proposes a new stochastic Polyak stepsize for stochastic non-smooth optimization, along with convergence rates under different assumptions.

**Strengths:**

The novel stepsize appears to slightly relax previous restrictions, though it comes at a new cost of needing to know $\ell\_i^\*$.

**Weaknesses:**

The most apparent drawback of this work is that its convergence is limited to a neighborhood the size of which is determined by the quality of estimation for $\ell\_i^\*$. The authors do not adequately address the issue of estimating $\ell\_i^\*$; see questions. Additionally, the statement their "safeguarded momentum rule removes this oracle dependence while preserving the same rate" is incorrect. The same rate is preserved only when the oracle dependence is the same as before i.e. $f\_i(x^\*) = \ell\_i^\*$. Otherwise, the rate has an extra $\sigma^2$ term, which makes it different.

The work also notes it "removes the need for any oracle information," that the approach is "fully adaptive, i.e, need no additional problem knowledge," and that it converges "without extra information." These statements are in conflict with the fact that $\ell\_i^\*$ are required---and $\ell\_i^\*$ must be such that $\ell\_i^\* \leq f\_i^\*$---which constitutes both "extra information" and "additional problem knowledge".

There is also the misleading statement that "using the safeguarded step size SPSsafe, we achieve the same $O(T^{−1/2})$ convergence to a neighborhood of the solution." You do not achieve the same $O(T^{−1/2})$ convergence as in previous works. You achieve $O(T^{−1/2})$ convergence to a neighborhood of the solution. These are not the same, and the authors should clarify this point throughout the paper. Similarly misleading statements are:

- "it needs neither the interpolation condition nor oracle access to the values $f\_i(x^\*)$ and still achieves the rate $O(T^{−1/2})$."
- "Our results are the first Polyak-type algorithm that converges to convex and non-smooth settings without extra assumptions"
- "providing convergence rate $O(T^{−1/2})$ for stochastic, convex–Lipschitz objectives, the first Polyak rule to do so without assuming interpolation or the knowledge of $f\_i(x^\*)$."

In all of these cases, proper qualification of converging to a neighborhood is required.

**Questions:**

- The authors note that "many modern machine learning models satisfy this condition" for interpolation, where $\ell\_i^\* = f\_i^\*$ implies $\sigma^2 = 0$. But this case was already handled in Loizou et al. (2021). Can the authors provide an example that does **not** satisfy the interpolation condition, and also 1) give values $\ell\_i^\*$ guaranteed to be lower bounds for $f\_i$ (with proof) and 2) give an upper bound on $\sigma^2$?

- The paper claims their "proposed analysis provides the first convergence guarantees for an adaptive momentum method (through the equivalence of IMA and SSM with heavy ball momentum) that does not require any strong assumption." What about the fact that the stepsize depends on $\ell\_i^\*$? Is this not a strong assumption? It also claims to "remove the need for any oracle information." What about $\ell\_i^\*$?

- Is it possible to explain, and substantiate, the extent of their contributions beyond generalizing Theorem C.1 in Loizou et al. (2021) to the non-interpolated setting, using the proof techniques from the rest of Loizou et al. (2021) (which Loizou et al. observe, in Appendix C.1, that "one can easily obtain" "using the proof techniques from the rest of" their paper)?

- What hyperparameter tuning was done for Adam baseline? What was the choice of stepsize?

---

> ### Author Response · Authors · 2025-11-20
> **Author's Rebuttal (Pt 1)**
>
> **We thank the reviewer for their detailed comments.**
>
> The main concerns of the reviewer are the (i) convergence to a neighborhood and (ii) the use of $\ell_i^\ast$ in the step-size selection. Before we address individual questions and weaknesses, let us clarify the main criticism of these two points.
>
> **Regarding the convergence to a neighborhood.**
>
> The reviewer claims that it is "misleading" to say that SPS_safe​ "achieves the rate $O(T^{-1/2})$" because we converge to a neighborhood. Indeed, we agree that our step-size selection guarantees convergence to a neighborhood of the solution. We precisely mention that in several parts of our original submission, and we politely disagree with the claim that our claims are misleading. Let us provide examples of statements in our original submission that made this clear:  See line 129 "for convex, Lipschitz objectives (up to a neighborhood) without extra information", line 138 "attains $O(T^{-1/2})$ convergence a neighborhood of solution, for stochastic, convex & Lipschitz objectives without the interpolation assumption", line 212 "convergence to a neighborhood of the solution", line 370 "Similar remarks as in the previous section  hold about the neighborhood in this regime", line 419 "a larger $M$ tightens the neighborhood: both algorithms converge to progressively lower error plateaus as the safeguard increases".
>
> Our results and the way we phrase them are fully in line with standard practice in the stochastic optimization literature. The additive neighborhood term $O(\sigma^2)$ comes from stochastic noise and appears in virtually all stochastic methods, including classical SGD with constant step-sizes [Gower et al; 2019, Garrigos and Gower; 2023], momentum variants [Liu et al; 2020, Sebbouh et al; 2021], and Polyak-type step sizes [Loizou et al; 2021, Oikonomou and Loizou; 2025]. Removing this term without assuming interpolation or other strong conditions typically requires either decreasing step sizes or variance-reduction techniques, both important but orthogonal directions to the main scope of this work. Finally, we emphasize that our guarantees strictly extend previous ones: under interpolation, we also recover $O(T^{-1/2})$ convergence to the exact solution (see Corollary 3.2).
>
> Having said the above, we see the point of further including the word neighborhood in the instances the reviewers mention (to avoid potential misunderstanding). In the updated version of our submission, we already did that.
>
>
> **On knowledge of $\ell_i^\ast$.**
>
> The reviewer claims that our statements on "fully adaptive, i.e, need no additional problem knowledge" and "removes the need for any oracle information" are in conflict with the fact that the lower bound $\ell_i^\ast$ are required, which constitutes both "extra information" and "additional problem knowledge".
>
> Here we highlight that the knowledge of $\ell_i^\ast$ is much more relaxed than both $f_i^\ast=\inf f_i$ and definitely $f_i(x^\ast)$ (see also Orvieto et al. 2022 for detailed arguments about this). The $f_i(x^\ast)$ needs to be part of the optimization oracle in the previous works for one to be able to run their step size, and it is only computable in the interpolation regime. As we mentioned in our paper, in most, if not all, popular ML applications, $f_i$ are non-negative functions, and simply using  $\ell_i^\ast=0$ is sufficient. We could have simply introduced the step-size without the $\ell_i^\ast$ by simply having $0$ instead in their place and have, for example, $\gamma_t=\frac{f_i(x^t)}{\max\{\|g_i^t\|^2,M\}}$ as SPS_safe. In that scenario, the convergence analysis will be identical to what we have with an updated $\sigma$ that depends only on the function values at the optimum. We note that in all of our experiments, we use by default $\ell_i^\ast=0$ without any modification, and show a great practical performance of the proposed algorithm. This is standard practice in any Polyak-type step-size in modern ML (use of zero instead of $f_i^\ast$ or $\ell_i^\ast$). With this in mind, we hope the reviewer is able to see that our statements on "fully adaptive, i.e, need no additional problem knowledge" and "removes the need for any oracle information" are accurate. In practice, the only requirement is the $f_i$ to be non-negative functions (even if this is not forced by our theory - theory holds under more relaxed conditions), which, for all ML applications we care about, is always true.
>
> *References:*
>
> Gower, Robert Mansel, et al.: SGD: General analysis and improved rates
>
> Garrigos, Guillaume, and Robert M. Gower: Handbook of convergence theorems for (stochastic) gradient methods
>
> Sebbouh, Othmane, Robert M. Gower, and Aaron Defazio: Almost sure convergence rates for stochastic gradient descent and stochastic heavy ball
>
> Yanli Liu, Yuan Gao, Wotao Yin: An Improved Analysis of Stochastic Gradient Descent with Momentum

---

> > ### Author Response · Authors · 2025-11-20
> > **Author's Rebuttal (Pt 2)**
> >
> > **Answers to the Questions.**
> >
> > **Q1:** Multiple examples satisfy our conditions. For example, let $f(x) = \frac{1}{2}\left[f_1(x)+f_2(x)\right]$ with $f_1(x)=(x-1)^2$ and $f_2(x)=(x+1)^2+1$. Then $x^\ast=0$, $f_1^\ast=0$ and $f_2^\ast=1$, thus $f_2(x^\ast)=2\neq 1=f_2^\ast$, meaning that **interpolation is not satisfied**. Note that it is obvious that $f_1(x)\geq0$ and $f_2(x)\geq1$, so we can choose $\ell_1^\ast=\ell_2^\ast=\ell^\ast=-1$, hence $\ell_i^\ast$ is **guaranteed to be lower bounds** for $f_i^\ast$ (with proof). Finally, we have $f_1(x^\ast)=1$, $f_2(x^\ast)=2$ and $\ell_1^\ast=\ell_2^\ast=\ell^\ast=-1$, so it’s easy to calculate $\sigma^2=\sqrt{13/2}$, so not only we give **an upper bound** on $\sigma^2$, we actually calculate it.
> >
> > **Q2:** We do not consider the knowledge of $\ell_i^\ast$​ (a lower bound on $f_i$) to be a strong assumption. Knowing the exact infimum $f_i^\ast$​ is substantially more demanding than knowing any valid lower bound. In practice, most standard ML losses are nonnegative, so one can simply take $\ell_i^\ast=0$, which is known a priori. Note that prior work (e.g., Loizou et al., 2021) effectively made the stronger assumption by using  $f_i^\ast=0$ in experiments, without explicitly justifying why the infimum should equal zero. In our setting, this choice is rigorously justified, since our theory only requires a lower bound and not the exact optimal value.
> >
> > **Q3:**
> > Even though Loizou et al. state that "using the proof techniques from the rest of the paper, one can easily obtain convergence for the more general setting" they actually provide guarantees only in the much simpler interpolated regime. Extending these results to the non-interpolated case is far from a routine exercise, particularly with the proof techniques used in that work.
> > In most previous stochastic Polyak stepsizes papers (see [Loizou et al, 2021], [Orvieto et, 2022]) the proofs start by expanding the quantity $|x^{t+1}-x^\ast|^2$, use convexity to upper bound it and then use inequalities on the stepsize $\gamma_t$ to further upper bound the expressions and finally arrive to a convergence result. This misses much of the structure of a Polyak-type stepsize. On the other hand, the original deterministic Polyak stepsize uses the exact expression for $\gamma$ to simplify the upper bound on $|x^{t+1}-x^\ast|^2$ based on convexity, thereby fully exploiting the Polyak stepsize. Very similar techniques are used in [Garrigos et al, 2023] and [Gower et al, 2025], and this is the reason they both need knowledge of $f_i(x^\ast)$. In our case, we have reverse-engineered the stepsize so that we are forced to use its precise expression on the upper bound. Then, a few algebraic tricks, as well as Cauchy-Schwarz and Jensen’s inequality, have been used to conclude the proof. Based on this, our proof techniques as well as our step-size selection differ substantially from previous Polyak-type ideas.
> >
> > **Q4:** The learning rate of Adam in our experiments is set to lr=0.001 (a choice we notice works well for all settings we focused on).
> >
> > **If you agree that we managed to address all issues, please consider raising your mark to support our work. From our viewpoint, all concerns raised (such as the neighborhood of convergence and knowledge of $\ell_i^\ast$) are mainly clarifications that have already been addressed in the updated version of our work, and not reasons justifying a rejection score.  If you believe this is not the case, please let us know so that we have a chance to respond.**

---

### Meta-Review · Area_Chair_7zhZ · 2025-12-26

**Summary:**

This work considers modifying the common Polyak step-size rules so that the denominator in the step-size expression is not too small. More concretely, the usual $\\| g_i^t \\|^2$ is replaced with $\max ( \\| g_i^t \\|^2, M )$, where $M>0$ is a user-specified parameter, and $g_i^t$ is a stochastic gradient for a loss-function component $i$ randomly chosen at iteration $t$.

There are major concerns in this paper that are not fully clarified:

- Reviewer o9nw has concerns about a few places in the writing, where the description of the results and contributions of the proposed method appears to be imprecise. After delving into this, I can see where these concerns are coming from. I believe it would be beneficial if this work were more explicit about its potential limitations and more accurate when describing its contributions. Unfortunately, the authors did not make a significant effort to revise the paper to incorporate this feedback, and thus the concerns are still not clarified. Reviewer o9nw have pointed out a few specific places where these issues arise, and I repeat a couple of them here:
1. **Line 377:** "Our safeguarded momentum rule removes the oracle dependence while preserving the same rate."
Same rate for the same target or metric? Same rate under the same set of conditions? Same rate without any relaxed or inexact oracle dependence?
2. **Lines 153--156:** "Our proposed analysis provides the first convergence guarantees for an adaptive momentum method (through the equivalence of IMA and SSM with heavy-ball momentum) that does not require any strong assumption (e.g., the knowledge of $f_i(x^*)$)."
It would be helpful to specify exactly what assumptions the proposed method and analysis do require. This would make the description of the novelty and contributions of this work more accurate.

- In addition to the writing issues, both Reviewer 6THh and Reviewer d8h5 pointed out that the theory is developed only for convex optimization problems, while the experiments are conducted on synthetic convex functions and on ResNet-20/32 trained on CIFAR-10/100. This shows a misalignment between the theoretical results and the empirical evidence. The work could be strengthened if a more diverse set of experiments on convex optimization problems were provided (particularly in settings where estimating a lower bound of the optimal value becomes a critical issue) to better support the theoretical analysis.

Finally, the authors might want to discuss the last component in the upper bound in their main theorems (Theorem 3.1 & 3.4) and check whether the bound is vacuous or meaningless in certain regimes. For example, when $G^2 > M$, where $M$ is the user-specified parameter, the authors might want to check whether the second component $\sqrt{ \frac{ \max( G^2, M) }{ M } } \sigma^2 $ can amplify the noise $\sigma^2$ significantly and also discuss its limitations if any.

**Reviewer Concerns:**

See above _Summary_ for the major concerns that are still lingering.

There were questions raised by Reviewer o9nw and Reviewer Zq5v regarding how the hyper-parameters were specified; the authors' replies have provided additional information about the experimental settings accordingly.

**Reviewer Scores:**

- Reviewer o9nW is unlikely to change the score, given that there is a significant gap between the reviewer and the authors on the presentation.

- Reviewer 6THh and Reviewer Zq5v might not champion this paper after reviewing the lingering concerns.

- Reviewer d8h5's concerns (on the alignment between theory and experiments) are not fully clarified in the rebuttal phase, and they are unlikely to upgrade their score given their high confidence score initially.

---

### Decision · Program_Chairs · 2026-01-26

Reject